# Population Dynamics of Spotted Wing Drosophila (*Drosophila suzukii* (Matsumura)) in Maine Wild Blueberry (*Vaccinium angustifolium* Aiton)

**DOI:** 10.3390/insects10070205

**Published:** 2019-07-13

**Authors:** Francis Drummond, Elissa Ballman, Judith Collins

**Affiliations:** 1School of Biology and Ecology, University of Maine, Orono, ME 04469, USA; 2Cooperative Extension, University of Maine, Orono, ME 04469, USA

**Keywords:** predation, sex ratio, action threshold, pest management, insecticide use

## Abstract

A long-term investigation of *D. suzukii* dynamics in wild blueberry fields from 2012–2018 demonstrates relative abundance is either still increasing or exhibiting periodicity seven years after the initial invasion. Relative abundance is determined by physiological date of first detection and air temperatures the previous winter. Date of first detection of flies does not determine date of fruit infestation. The level of fruit infestation is determined by year, fly pressure, and insecticide application frequency. Frequency of insecticide application is determined by production system. Non-crop wild fruit and predation influences fly pressure; increased wild fruit abundance results in increased fly pressure. Increased predation rate reduces fly pressure, but only at high abundance of flies, or when high levels of wild fruit are present along field edges. Male sex ratio might be declining over the seven years. Action thresholds were developed from samples of 92 fields from 2012–2017 that related cumulative adult male trap capture to the following week likelihood of fruit infestation. A two-parameter gamma density function describing this probability was used to develop a risk-based gradient action threshold system. The action thresholds were validated from 2016–2018 in 35 fields and were shown to work well in two of three years (2016 and 2017).

## 1. Introduction

The spotted wing Drosophila (SWD), *Drosophila suzukii* (Matsumura) (Diptera: Drosophilidae), is an invasive vinegar fly unintentionally introduced from Asia. It was first documented in the continental United States in 2008 [1]. After its introduction, *D. suzukii* spread rapidly across much of the country including Maine where it was first documented in 2012 [2]. Unlike native *Drosophila*, the females have a serrated ovipositor that allows them to oviposit into intact ripe fruit. The larvae develop inside the fruit. This causes softening and introduces microbes that shorten shelf life and renders fruit unmarketable [3]. The attack of undamaged ripe fruit along with its wide host range has made it a serious insect pest in many fruit crops such as cherries, raspberries, blackberries, strawberries, and blueberries [4,5].

Maine produces 10% of all blueberries in North America [6]. Maine’s wild blueberry (lowbush) crop (*Vaccinium angustifolium* Aiton) is produced on approximately 17,000 ha of managed land. Production contributes over $250 million to the state’s economy each year [6]. Wild blueberries are unique in that they are grown on a two-year cycle; the first year is a prune year and is characterized by vegetative growth and the development of flower buds. The second year is when bloom and berry harvest occur [7]. Growers in Maine produce wild blueberries both organically and conventionally. There are three levels of conventional production (low, medium, and high input) [8,9]. These production systems vary in the intensity of capital inputs used to produce the crop. Conventional growers rely primarily on insecticide based spotted wing drosophila pest management; there are fewer options for organic control of *D. suzukii* [10]. *Drosophila suzukii* caused an estimated $1.4 million crop loss in Maine wild blueberry during 2012, the first growing season it was present in the state [11]. Falling blueberry prices coupled with increased production costs for monitoring and managing *D. suzukii* have presented challenges to Maine’s blueberry growers [7]. 

Since its first detection in the continental United States in 2008, much research has been conducted, though there are still gaps in knowledge that limit our understanding of this pest. To date, most studies of *D. suzukii* population dynamics have only investigated population fluctuations across one, and sometimes, two years [12,13,14,15]. One study, Briem et al. [16], analyzed adult trap catch data in Germany from 2012–2018. Except for their study, data are lacking on long-term population dynamics immediately following the invasion of this species. Invasive species population dynamics can change after their initial establishment with short-term population dynamics quite different from long-term population dynamics [17,18]. There have been several investigations that have evaluated factors appearing to affect population growth and immature and adult survival. Predation and parasitism of immature stages by natural enemies has been investigated both in Europe and North America [19,20,21,22] and in its native range [23,24]. Winter temperatures appear to affect overwintering survival of adults [25,26,27,28,29] and extreme high lethal temperatures have been shown to reduce longevity of adults during the summer [30,31,32]. In addition, dispersal over long distances likely maintain meta-populations of *D. suzukii* [33]. Landscape level plant resources for reproduction and intrinsic rate of growth have also been shown to affect population levels [5,34,35,36,37,38,39].

In order for growers to respond effectively to continued *D. suzukii* threats, they and pest management specialists need to understand factors that affect population increase in this invasive pest. Our study examined population trends of *D. suzukii* in Maine wild blueberries over the course of seven years. Studying *D. suzukii* over multiple years may help growers know what they can expect from this invasive insect population so they can make appropriate decisions on monitoring and management. This study is organized into the following topics: (1) annual trends and phenology, (2) effects of management on adults and fruit infestation, (3) effects of wild fruits and natural enemies, and (4) action thresholds for predicting fruit infestation and grower decision-making. We measured relative abundance of adults, the first date of *D. suzukii* adult appearance and fruit infestation each year, sex ratio and the relationship between adult density and fruit infestation levels across all seven years. We investigated potential relationships between population dynamics and weather and what impact different management systems and production stages (organic versus conventional, vegetative versus fruiting fields, and pesticide usage) have on *D. suzukii* populations. Our hope is that this research provides growers with a better basis for development of integrated pest management (IPM) tools for monitoring, predicting, and controlling *D. suzukii*, especially the practicality of action thresholds. 

## 2. Materials and Methods 

### 2.1. Data Collection and Analysis Software

#### 2.1.1. Sampling Adult *D. suzukii* by Trapping to Estimate Population Relative Abundance 

This study was conducted over a period of seven years (2012–2018) in Maine wild blueberry fields to investigate the population dynamics of D. suzukii. Field studies were conducted in 10–20 managed wild blueberry fields per year in Hancock, Knox, Lincoln, Waldo, and Washington counties, Maine, USA (Table 1). Baited traps were used to estimate the relative abundance of adult fly populations in wild blueberry fields. All adult sampling between 2012–2018 utilized spotted wing drosophila traps consisting of 473 ml of red polystyrene Solo^®^ (Dart Container Corp., Mason, MI, USA) cups with 7–10, 0.48-cm-diameter holes punched evenly around the top rim and light-blocking lids [2,40]. Traps were baited with 118-mL standard yeast/sugar bait (2012–2018) (5.07 g dry active yeast, 25.35 g sugar, 450 mL water) [2,40]. This bait is highly attractive to *D. suzukii* in the Maine wild blueberry landscape [40,41]. For all trials in 2012, a yellow sticky card was also hung inside each trap to facilitate captured fly species identification. In all sampling (2012–2018) traps were hung 0.30 to 0.61 m above the top of the blueberry canopy using 0.91-m plant stands and were placed along wild blueberry crop field edges. The exception to this trap placement protocol was for the crop cycle study, described in Section 2.5 where traps were deployed along the edges of both crop and vegetative fields. Traps were deployed at densities of 3–5 traps per field (more traps in larger size fields) and checked 1–2 times per week. At each visit, traps were replaced with new traps and fresh bait. Traps that were removed from the field were returned to the lab and drained through a fine mesh sieve to count the number of male and female *D. suzukii* [2]. 

#### 2.1.2. Sampling Larval Infestation

Wild blueberry infestation by larvae was determined by visiting fields weekly, once fruit was vulnerable to larval infestation, the first signs of individual ripe fruit [2]. Blueberries were sampled by collecting 1 L samples (ca. 500 berries/sample) at different locations (blueberry clones) representatively throughout the field. The berries were stored in a cooler with an ice pack for transfer to the laboratory. In the laboratory, fruit was added to a 4 L Zip Lock^®^ (S. C. Johnson & Son, Inc., Racine, WI, USA) polyethylene bag. A 1-L 10% saline solution was mixed (1 part salt:10 parts water, v:v) and ca. 236 mL was added to the bag containing the berries. The bag was sealed and the berries were gently pressed with a wooden block just enough to crack the skins. The split berries were allowed to remain in saline for 60 min. The berries and saline liquid were then strained through a coarse sieve into a dark colored tray. The fruit in the sieve was discarded and the strained liquid was inspected for larvae that sink to the bottom of the pan (modified from Drummond et al. [2]). 

#### 2.1.3. Statistical Analysis and Modeling 

We used Microsoft Excel^®^ for database management. The statistical software JMP PRO version 14 [42] was used for all analysis. The specific hypotheses, approaches and statistical models are mentioned under each subsection below. Bonferroni corrected P-values were estimated when multiple sequential tests were performed. Kaliedagraph^®^ version 4.5.1 graphical software [43] was used for non-linear least squares (Levenberg-Marquardt), and the minimization of sums of squares was based upon the partial derivative solutions developed by one of the authors, F. Drummond. 

### 2.2. Seasonal and Annual Adult D. suzukii Population Relative Abundances 

We conducted exploratory analyses to test the following hypotheses: (1) *D. suzukii* relative abundances have increased either linearly, non-linearly, or are periodic since the year of first arrival in Maine wild blueberry (2012); (2) rate of fly trap capture (on a per day basis) within the growing is geometric and variable among years; (3) the degree day of first adult trap capture has become earlier as the invasive *D. suzukii* population has become established over time; and (4) annual weather conditions determine *D. suzukii* fly abundance levels in commercial wild blueberry fields. 

In order to determine changes in seasonal occurrence in *D. suzukii* populations (2012–2018), we sampled adult population levels in wild blueberry fields using the trapping method previously described (Section 2.1.1). We refer to trap capture abundance of flies as relative abundance throughout this study because we realize that trap capture is only an index of absolute density as trap captures are affected by behavior, weather conditions, the habitat, and their interactions.

Across the seven years, using general linear models, we assessed *D. suzukii* seasonal rate of increase (based upon trap capture of adults), annual trend in adult relative abundance (based upon cumulative adult trap capture at degree day 705 (base 10 °C, with 1 April as the start date). This degree day index corresponds to average length of the wild blueberry growing season, 1 April to early September (harvest). Maximum and minimum daily air temperatures were used from the nearest weather station to each field and that the calculation of cumulative degree days was initiated on 1 April of each year. In order to calculate the number of degree days accumulated for each day, we added the maximum and minimum air temperatures for that day and divided by two to estimate the average temperature for that day ([Max + Min]/2 = Average Daily Temperature). Then the developmental threshold, in our case, 10 °C was subtracted from the daily average temperature to derive the number of degree days ([Average Daily Temperature − Developmental or Base Temperature] = Degree Days). If the resulting degree day estimate is equal to or greater than 0.0, then this number of degree days is added to the cumulative degree days to this point. If the number of degree days derived for that day is less than 0.0, then 0.0 is added to the cumulative degree days). Tochen et al. [30] suggested a developmental threshold of 7.2 °C for *D. suzukii*, but state that temperatures between 10–28 °C resulted in significant impact on blueberries and so we adopted the threshold of 10 °C.

We selected a “standardized growing season” degree day accumulation so that *D. suzukii* relative abundances could be compared between years. This as described above is the cumulative number of degree days typical in most years that encompasses the start of plant development in the spring (1 April) to harvest (mid-August) [44]. Therefore, this standardized growing season is a plant development period that is relative to *D. suzukii* activity. The rationale for selecting a standardized physiological time or standardized growing season is so that relative abundance, based upon trap capture, could be compared from one year to the next. Not all fields could be used in the standardized degree day comparison because some fields were too distant from a local weather station and some fields were harvested long before the accumulated degree days occurred. To test the hypotheses stated above we conducted the following analyses.

First, to determine if logarithm of annual fly captures per trap over the season (using standardized growing season of 705 DD as length of season) increased linearly or nonlinearly over time or were characterized by periodicity (2012–2018), we fit the following models for testing this hypothesis: (1) a linear generalized model (Gaussian error) using maximum likelihood and year as the independent variable; (2) non-linear generalized model, polynomial in form (fourth order) regression; (3) a penalized piece-wise spline regression (to prevent over-fitting of the data); and (4) a Fourier sine/cosine non-linear model [42]. The piece-wise spline and the Fourier models were fit to the time series data using a generalized mixed model where the spline or Fourier coefficients are treated as random effects estimated as best linear unbiased predictors [42]. Visual inspection of model residuals compared to model predictions and AICc (Akaike information criterion corrected for small sample sizes) were used to provide evidence for selection of a model that would best support either a linear increase or the non-linear alternative models representing periodic or oscillatory dynamics. In addition, we tested if the alternative non-linear models significantly reduced the unexplained variance in the base linear model by comparing the model deviances (−2 log likelihood statistics) of the non-linear models to the base linear model with the Chi-square distribution. 

Second, we used general linear models to determine if daily capture rates and cumulative daily capture during the growing season (logarithm transformed) varied among years. Estimates of annual rates of trap captures (slopes) were compared by assessing the overlap in 95% confidence intervals among the slopes years. 

Third, we determined if the cumulative degree day (initiated on 1 April) of first adult capture is occurring earlier in the growing season. We used a general linear model to test this hypothesis where year was the independent variable and degree day at first occurrence was the dependent variable. 

Fourth, we investigated potential weather factors that might predict logarithm transformed cumulative daily and total fly captures per trap prior to harvest. General linear models were constructed with independent variables of mean spring and early summer air temperatures (June–July), mean winter air temperatures prior to the growing season, and degree day of first fly capture.

### 2.3. Sex Ratio Changes during the Study

Sex ratio of fly trap captures was assessed over the seven year study using two methods. First the number of captured males vs female trap capture in each field in each year was assessed to determine if total male captures relative to total female trap captures were independent of relative abundance. This hypothesis was tested using linear regression of log transformed trap captures (log males vs log females). The slope was evaluated for significance and if it was different from 1.0. A second analysis was conducted to determine if sex ratio changed over the seven-year period of the study. Sex ratios (males/ (males + females)) were analyzed for every sampled field for each year. A binomial logistic regression a model with the number of males the successes and the total number of draws (the number of males plus the number of females) was used to test if sex ratio changed over time. Odds ratios were calculated by year and over the seven-year period to quantify the likelihoods of sex ratio change per year and over the entire seven-year study period.

### 2.4. Management System Impact on D. suzukii Relative Abundance and Fruit Infestation

In order to determine the response of *D. suzukii* to management system (2012–2017), fifty-eight fields were grouped by blueberry production system (organic or conventional: low, medium, or high input). Our hypothesis was that organic production systems would have higher average *D. suzukii* relative abundance and high input production systems would have lower relative abundance levels because of more intensive management. Typical levels of inputs used by growers in the four management systems for the production of wild blueberries are given in Table 2. The cumulative number of adult *D. suzukii* per trap in each field type was compared at degree day 705 and the degree day of first infestation using fixed model analysis of variances (general linear model) with production system (categorical) and year (continuous) as main effects along with their interaction. Poisson regression was used to determine if insecticide application frequency was determined by production system. Multiple paired independent Poisson contrasts (Bonferroni corrected) were used to determine which production systems differed in insecticide application from each other. 

We also investigated the distribution of the severity of larval infestation among fields by year using a general linear model with year as the independent variable. In order to describe the distribution of infestation rates for each sampled field over all years, an empirical frequency distribution was constructed. A theoretical probability density function could only be fit to the distribution of infestation rates (excluding non-infested fields). A Kolmogorov’s D test was used to determine the goodness of fit of the empirical data and the theoretical probability density function [42]. The impact of insecticides on larval infestation and adult relative abundance measured as the logarithm transformed cumulative adults captured per trap by standardized growing season cumulative degree day 705. In most fields during each year, once fruit became susceptible to attack (ripe), blueberries were hand harvested and inspected for larval infestation (for details see Section 2.1.2, Sampling larval infestation). Logistic regression was used to determine factors that predict blueberry infestation rate (proportion infested fruit) prior to harvest. The independent variables were logarithm (base 10) transformed cumulative fly abundance/trap, year, and the number of late summer insecticides (these are applied both to blueberry maggot fly, *Rhagoletis mendax* Curran (Diptera, Tephritidae), and *D. suzukii*, sometimes simultaneously) applied to the field. The number of insecticides used in fields that we sampled were recorded by us after interviewing each grower who agreed to provide this information (*n* = 86 fields between 2012 and 2018).

### 2.5. Crop Cycle Effect on D. suzukii Relative Abundance 

Wild blueberry production in almost all fields in Maine is on a two-year cycle. In the first year of the cycle plants grow vegetatively and do not produce fruit. During this time flower buds are produced that will bloom the following year. In the second year, bloom occurs and fruit is produced and harvested. Our hypothesis was that fields in the vegetative or prune cycle have lower relative abundance of *D. suzukii* than fields that bear fruit (crop fields). Ten fields were simultaneously sampled for adult *D. suzukii* relative abundance in vegetative vs fruiting fields (five fruiting and five vegetative fields, a pair at each of five geographic locations) in 2012. Three traps were hung in each field located in Columbia, Columbia Falls, and Jonesboro, Washington Co. ME on 11 August for one week. Analysis of variance (randomized complete block design, paired field location as block) was used to compare trap capture of *D. suzukii* adults in crop (fruit-bearing) versus prune (vegetative) fields. The dependent variables for three non-replicated two-way ANOVAs (randomized complete block designs) were male, female, and total fly captures per trap.

### 2.6. Impact of Natural Enemies and Wild Fruits on D. suzukii Relative Abundance in Wild Blueberry Fields

We have previously shown that spotted wing drosophila utilize wild non-blueberry fruits along field edges to build up their populations prior to blueberry ripening [39]. We have also shown that predation of SWD pupae in wild blueberry fields can be quite high and appears to be mostly associated with insect predators, especially crickets [21]. This study, conducted in 2018, was designed to assess how wild fruit utilization and predation affect the population buildup of *D. suzukii* in wild blueberry fields. Our hypothesis was that fields with high abundance of wild fruits on the periphery of wild blueberry fields would have high *D. suzukii* relative abundance and fields with high abundance of natural predators would have low relative abundance by the end of the growing season and that these dynamics occur independent of field size. Twenty commercial wild blueberry fields were selected in 2018, 10 in each of the two major growing regions (Midcoast and Downeast) as described in Section 2.1.1 and Table 1. Three traps were placed in each field in early July, 2018. At the time of trap deployment field size was obtained from the growers. Traps were monitored at 5–7 day intervals until harvest. Throughout the study, on each sample date, traps set the previous week were collected and returned to the laboratory where male, female, and total abundance adults were determined and recorded. New traps were deployed weekly. Using these data, we calculated the total number of *D. suzukii* adults and males per trap captured from each site for each date and the mean cumulative number of adults and males over the entire sampling period. 

An index of predation relied upon deploying sentinel *D. suzukii* pupae in the fields (see Ballman et al. [21] for detailed methods). Pupae were removed from our laboratory colony, rinsed under running water to remove media, and examined under magnification to verify the presence of a developing fly. All pupae were frozen for 24 h prior to the start of the experiment. Killed pupae were affixed to 9 cm white-painted Petri dishes by two, 7 cm rows of double-sided tape. Each piece of tape had 10 pupae for a total of 20 pupae per plate. At each field site, three plates were placed along the field edge 3 m apart and lightly covered with duff collected from the field. All plates were left in the field for 24 hours and then retrieved. The numbers of intact pupae were counted in the lab under magnification. Pupae that were obviously chewed, but not completely consumed, were counted as predated.

Wild fruit host abundances were quantified along the edges of wild blueberry fields. Wild fruit was surveyed 2–3 times in blueberry fields in late July, early August, and late August. To evaluate wild fruit, three, 30-m transects were set up along each of four field edges. The presence or absence of wild fruit was recorded at every other 0.3 m along each transect for a total of 50 observations per transect. The species of wild fruit along each field were also recorded (see Ballman et al. [21] for more details). 

We constructed a general linear model to determine the field level factors that explained the variation in the end of season total (male + female) *D. suzukii* captures per trap in each field. The potential predictors that were considered for the model were: field size (ha), production system (organic, and low, medium, and high conventional), the prevalence (% of transect landcover) of non-crop wild fruit along the edges of the field, and the intensity (% of sentinel pupae predated) of predation on pupae in the field. In addition, we included all of the two- and three-way interactions in the model. 

### 2.7. Management-Early Harvest Tactic and Action Thresholds for D. suzukii in Wild Blueberry

The wild blueberry fields sampled in the seasonal occurrence study described above (Section 2.1) were used to determine if action thresholds for *D. suzukii* adults could be developed for growers. Our approach was to base the action thresholds on adult male trap captures. Once pigmentation develops in the wings of newly emerged males, they are easy to identify with little error by growers in Maine. There are no other species of drosophila or other small sized Diptera in Maine that have similar wing pigmentation and can be confused with male *D. suzukii*. A statistical model to determine the relationship between male trap capture and percent fruit infestation the following week was fit to all wild blueberry fields that were sampled for infestation prior to harvest (*n* = 92) between 2012 and 2017. Thus, fields not sampled for infestation of fruit before harvest could not be used for this model even if they were monitored for adult relative abundance. Because the relationship was non-linear in nature, we fit a linearized exponential model to the data to determine if a significant relationship existed and then using the Levenberg-Marquardt’s non-linear least squares algorithm, we fit a three-parameter exponential model to the data to construct a predictive equation [45] (see Methods, Section 2.1.3: Statistical Analysis and Modeling).

We first developed a tactic of early harvest. In all years (2012–2018) fields that were harvested prior to any *D. suzukii* males being captured were assessed as to their level of infested fruit and the date of harvest. From this data (*n* = 10), we hypothesized that an early harvest tactic to avoiding damage was practical. To test this hypothesis, the proportion of non-ripe fruit present in 11 fields was assessed at harvest in 2016 in order to show the cost–benefit of an early harvest tactic. Three samples of 500 fruit throughout each field were collected in order to determine the mean and standard error of percent ripe fruit (green fruit/(green + ripe fruit)). A predictive linear model was developed to determine the crop loss due to non-ripe fruit and the time of harvest. 

Our second approach was to develop “risk-based” action thresholds. This involved fitting a probability density function to all fields (*n* = 92). The density function was then used to determine the probability or likelihood of infested fruit the week following a given mean cumulative male adult trap capture. We tested several continuous variate density functions as models for our empirical data. Depending upon the density function, a Kolmogorov-Smirnov cumulative frequency distribution test or a Cramer-von Mises frequency distribution test was used to evaluate the goodness of fit of the selected density function to the empirical data (α = 0.05). This information was then used to construct action thresholds for growers with varying risk aversion, based upon male *D. suzukii* in trap captures. Validation of the early harvest tactic and action thresholds derived from the probability model was performed on 14, 10, and 19 fields in 2016–2018. The 2016 validation was upon a probability model derived from a subset of fields sampled between 2012 and 2016 (*n* = 82); whereas, the 2017 and 2018 validations were based upon the probability model derived from fields sampled between 2012 and 2017 (*n* = 92). Fields were pre-assigned with an action threshold. The fields were monitored for the growers and with the assigned thresholds in mind, fields were either harvested before or immediately after the threshold was reached, or the growers applied an insecticide. The observed infestation rates of fruit for each of the action thresholds attained the previous week were then compared to the expected probability of infested fruit derived from the probability density function of male *D. suzukii* action thresholds.

## 3. Results

### 3.1. Seasonal and Annual Adult D. suzukii Population Relative Abundances

A linear model of the cumulative trap capture of *D. suzukii* flies (logarithm transformed) at a standardized growing season of degree-day 705 over the seven year period (*F*_(1,95)_ = 4.602, *p* = 0.035, *r*^2^ = 0.046) is shown in Figure 1. A four parameter Fourier model comprised of sine and cosine coefficients was the best of the alternative models at reducing the deviance (−2 log likelihood statistic) compared to the linear model (*p* < 0.001). The AICc was 235.4 for the linear model and 193.9 for the Fourier model. Figure 1 shows the predicted model for both the linear and Fourier model. While the model residuals appear visually to be better suited for the Fourier model, a paired T-test showed no evidence (*p* > 0.05) to suggest significant differences between the two model residuals, although the residuals were larger on average for the linear model. The linear model suggests a steadily increasing fly relative abundance over the seven-year period, while the Fourier model suggests a periodic fluctuating dynamic.

Figure 2 illustrates the logarithmic cumulative increase in trap captures per day over the growing season of *D. suzukii* in all the wild blueberry fields sampled and pooled for each year between 2012 and 2018. The increasing logarithmic trends were significant (*F*_(13,1101)_ = 658.42, *p* < 0.0001, *r*^2^ = 0.642), but a year by date interaction exists (*p* < 0.0001), suggesting that the rates of increase varied by year. When assessed among years (Figure 2), 2013 and 2014 rates of increase were less than 2018 which was the highest rate of increase, although not significantly different than 2012, and 2015–2017. 

We also calculated the rate of daily *D. suzukii* trap capture increase for a set amount of physiological time (proportion daily increase from first detection to the degree day standardized growing season) in each field. The average daily trap capture rate of increase for 2012–2018 was 2.15 ± 0.48 (s.e.). The highest daily rate of increase was in 2017 at 8.70 ± 2.52. The factors that were significant predictors of rate of trap catch were year (*p* < 0.0001) and spring degree day accumulation (May and June, *p* = 0.004). The overall model explained 48.7% of the variance in rate of increase in trap captures (*F*_(1,86)_ = 19.786, *p* < 0.0001). However, the slope of spring degree day accumulation was negative (*β* = −0.004 ± 0.001), which we cannot explain. Therefore, spring degree day accumulation might be negatively correlated with a predictor that we did not model.

We found that over the seven year invasion period, first captures of flies occurred earlier, measured in degree days, each year (*F*_(1, 99)_ = 19.939, *p* < 0.0001, *r*
^2^ = 0.168). This decline has a slope of −82.8 degree-days per year. The trend in the degree days at first detection of adults between 2012 and 2018 is shown in Figure 3.

Prediction of log cumulative fly capture/trap at the standardized growing season date was significantly dependent upon the degree day of first fly trap capture (*F*_(1,86)_ = 115.807, *p* < 0.0001, *β* = −0.003) and the mean winter air temperature prior to the growing season (*F*_(1,86)_ = 6.097, *p* = 0.027, *β* = 0.012). Year was not used in the model because we wanted to test annual weather effects. The overall model explained 63.6% of the variance in log cumulative fly capture per trap at the standardized growing season of 705 degree days (*F*_(2,86)_ = 58.656, *p* < 0.0001, *r*^2^ = 0.636). We sought to determine if year to year variation in growing season accumulated degree-days for the months June–August could explain cumulative *D. suzukii* trap capture at the standardized growing season of degree-day 705, but there was no relationship (*p* = 0.142).

### 3.2. Sex Ratio Changes during the Study

We found that the slope of the regression relating log (female flies/trap) to log (male flies/trap) in each field suggested that the sex ratio is constant across fly population density (*F*_(1,107)_ = 440.271, *p* < 0.0001, *r*^2^ = 0.815, *β* = +1.065, Figure 4A) across the seven-year period. Figure 4A also suggests that relative abundances of sexes is slightly male biased. Sex ratio (males/(males + females)) declined in a linear fashion between 2012 and 2018 (χ^2^_(1)_ = 10239.6, *p* < 0.0001, Figure 4B). The odds ratio of the change in sex ratio per year is estimated at 0.713, and over the entire period between 2012 and 2018 is 0.131. These odds are significantly different than 1.0 (*p* < 0.05), and suggest both an annual average and a seven-year period decline in male sex ratio. 

### 3.3. Management System Impact on D. suzukii Relative Abundance and Fruit Infestation

Many fields had a low percentage of infested fruit prior to harvest. Figure 5A depicts the frequency distribution of fields with infested fruit over the seven-year study. An Exponential probability density function λ = 1.108 ± 0.192 [s.e.]) was a good fit for the empirical data of infestation rates, excluding fields were no infestation was detected. The level of infestation by year can be seen in Figure 5B. There was been no observed trend in either increasing or decreasing infestation rates over time (*p* = 0.864, sin^−1^ √proportion transformed). Unlike first fly capture, we did not find any evidence to suggest that infestation of fruit occurred earlier, with respect to degree days, over time (*p* = 0.667). Logistic regression provided evidence that year (χ^2^_(6)_ = 15.859, *p* = 0.017, unit odds ratio among years: range of pairwise comparisons among years = 0.001 - 85.599), log (cumulative fly capture/trap) (χ^2^_(1)_ = 60.559, *p* < 0.0001, unit odds ratio = 1630.4), and the number of summer insecticide applications (χ_(1)_ = 4.553, *p* = 0.047, unit odds ratio = 0.278), were all significant predictors of the proportion of blueberries in the field infested prior to harvest. The coefficient for the logistic regression estimate of percent fruit infestation as a function of insecticide application was: −1.281 ± 0.271. This suggests that insecticide applications determine fruit infestation. However, inspecting the odds ratios for both the independent variables insecticide applications and log (cumulative fly capture/trap), it can be seen that the most influential predictor is the log (cumulative fly capture/trap); year is sometimes highly influential and at other times not, and insecticide applications are influential, but not as much as log (cumulative fly capture/trap), or some year effects. Year in our model may represent weather effects. Figure 5C shows the relationship between the estimated probability of blueberry infestation and the number of insecticide applications (logistic regression). 

We found that management system did not directly determine the logarithm of cumulative relative abundance of flies per trap (*p* > 0.05), nor the date of first fly detection. However, we have just shown that insecticide applications affect both logarithm of cumulative relative abundance of flies per trap and fruit infestation rate. Poisson regression suggested that wild blueberry management system does determine the frequency of pesticide applications in a model that includes both year and management system (χ^2^_(8)_ = 45.551, *p* < 0.0001). Organic systems used significantly fewer insecticide applications than the three conventional systems (low, medium, and high). Figure 6 shows the results of individual Poisson contrasts separating the number of mean insecticides applied in each management system. Therefore, management system does affect relative abundance of flies per trap and fruit infestation rate, but indirectly through insecticide application frequency which varies both within and between management systems. 

### 3.4. Crop Cycle Effect on D. suzukii Relative Abundance

We found that crop cycle determines adult relative abundance/trap. Significantly fewer males, females, and total flies were trapped in vegetative fields compared to fruiting fields (*F*_(1,8)_ = 6.346, *p* = 0.036; *F*_(1,8)_ = 7.803, *p* = 0.023; *F*_(1,8)_ = 6.905, *p* = 0.030; for males, females, and total flies respectively). These results are shown in Figure 7.

### 3.5. Impact of Natural Enemies and Wild Fruits on D. suzukii Relative Abundance in Wild Blueberry Fields 

We found that field size and production system did not account for a significant proportion of the variation in adult trap capture. Predation and wild fruit hosts were significant predictors, resulting in 53.9% of the variation in the total *D. suzukii* adult trap capture relative abundance across all fields (F_(3,16)_ = 6.258, p = 0.005, r^2^ = 0.539). Table 3 lists the model coefficients.

An inspection of Table 3 suggests that the two predictors do not operate independently, in fact, % pupal predation is only of value in explaining relative abundance in the context of the alternative wild fruit abundance along the edge of a blueberry field. The abundance of wild fruit is a strong predictor on its own (explaining 39.5% of the variation in *D. suzukii* relative abundance). Figure 8 depicts this interaction between the two predictors as a thermal map. Inspection of this figure shows that the observed adult relative abundance increases from low (dark blue) to high (bright red) as the % wild fruit in a blueberry field increases (left to right of graph). However, an increase in % predation (bottom to top of graph) only results in a decrease in *D. suzukii* relative abundance when the percent wild fruit abundance is high or greater than 50%. This is the significant model interaction between % pupal predation and % wild fruit: percent predation only appears to be influential and reduce *D. suzukii* relative abundance in fields with high levels of wild fruit. These fields also have higher adult relative abundances.

### 3.6. Management - Early Harvest Tactic and Action Thresholds for D. suzukii in Wild Blueberry

The relationship between male *D. suzukii* cumulative trap capture (logarithm transform) just prior to harvest and percent fruit infestation was significant and exponential in form (linear model: *F*_(3,89)_ = 88.044, *p* < 0.0001, *r*^2^ = 0.494). The coefficients of a non-linear fit exponential model of the form, % infestation = a + b × e^(c × male trap capture)^ were: a = −0.675, b = 0.445, and c = 1.129. The model fit and observed data points are shown in Figure 9A. 

The early harvest tactic has great promise for avoiding the consequences of fruit infestation due to *D. suzukii* attack. Table 4 shows that fields that are harvested prior to *D. suzukii* adults being captured in at least three traps per field are not likely to be characterized by infested blueberries. None of the fields that we monitored that were harvested early had any detectable fruit infestation (Table 4). However, we did find a cost to harvesting early. Figure 9B shows that non-ripe green fruit may more than offset the damage from *D. suzukii.* The relationship between date of harvest and the percentage of ripe (blue) berries harvested as a percent of the total berry yield is: % ripe fruit = 100/[1 + e^(30.903 − 0.159 × Julian Date)^], *p* = 0.002, *r*^2^ = 0.795).

A two-parameter Gamma probability density function best fit our empirical data comprising fields with mean numbers of cumulative male trap captures prior to harvest and the likelihood of not being infested the following week after the cumulative trap capture. Estimates for the two parameter Gamma distribution were: alpha (shape) = 1.6765 (1.0949−2.4512, 95% CI), and sigma (scale) = 9.43259 (6.1286–15.7678, 95% CI). The Cramer-von Mises W^2^ test suggests that the empirical data is not significantly different from the theoretical frequency distribution (W^2^ = 0.153, *p* = 0.25). Figure 9C shows the empirical frequency distribution and the theoretical model fit. Table 5 shows the risk-based action thresholds derived from the Gamma probability density function. The series of action thresholds allow growers with different levels of aversion to risk to select the threshold that fits their philosophy or financial vulnerability.

Validation of the thresholds from 2016–2018 is depicted in Table 6. It can be seen that in 2016 and 2017 the observed infested fruit levels associated with the selected action thresholds were similar to the expected model predicted levels. 

## 4. Discussion

Briem et al. [16] in a long-term trapping study cautioned about regarding trap capture as absolute population density. We agree that their warning is warranted. In a previous study we showed that only 15–20% of flies in a caged population are captured by the type of trap that we used in this study [46]. In addition, Briem et al. [16] show that trap captures for *D. suzukii* can be affected by the environment surrounding the trap and weather conditions. This has been shown for other insect species as well [47,48]. Therefore, in this study we refer to trap capture as a relative abundance estimate of *D. suzukii* and that trap captures only provide an index of population density. While we have observed its rate of trap capture to increase logarithmically, within a growing season, this is not related to the production system or the rate of degree day accumulation during the summer. Most crop production systems in other states also are characterized by explosive logarithmic rates of trap captures during the stage that ripe fruit is available for *D. suzukii* [14,31,41]. The rate of increase in trap capture during the growing season was explained by a negative relationship with degree day accumulation in the spring. It is not clear why cooler springs would result in a higher rate if trap capture increase in the late summer. We speculate that either the rate of spring degree day accumulation is negatively correlated to a causal factor that we did not measure or that it might be related to overwintered adult synchrony with spring available food resources such as early ripening fruit (ex. *Cornus canadensis* L. (bunchberry), *Lonicera* spp. (honeysuckles) and *Rubus* spp. (dewberry species) and/or early fungi that produce fruiting bodies. The authors have observed *D. suzukii* adults on the surface and larvae in the interior of fungal fruiting structures (toadstools) in Maine.

We found that the date of first detection determined cumulative relative abundance of flies at a standardized growing season of 705 DD. The earlier flies were captured in a field, the higher the relative abundance at this point in the growing season. Pelton et al. [13] found that in higher altitude woodlands, flies were detected earlier in the season than in low croplands, but unlike our study, they did not find any difference in fly relative abundance due to earlier detection. Briem et al. [16] found that the rate of degree day accumulation in the summer determined seasonal relative abundance of flies. We might be witnessing the same phenomenon, but it might be confounded by the increasingly earlier detection of flies in wild blueberry fields. 

Our hypothesis that *D. suzukii* relative abundance has been increasing each year is borne out by a linear model, but an alternative Fourier model suggests a periodic dynamic and not a simple linear increase. We base this on the better balance of residuals for the non-linear model (see Figure 1), reduction in model deviance and the better AICc index for a Fourier model compared to a linear model. However, we are uncertain about the conclusion of the *D. suzukii* relative abundance being periodic. A seven-year study is not a long enough time horizon to provide a confident conclusion. Only future monitoring of the population will allow more certainty. An annual oscillating relative abundance might suggest that predation is dampening *D. suzukii* population increase from one year to another. We have shown that predation can reach 100% of deployed sentinel *D. suzukii* pupae in commercial wild blueberry fields [21]. However, another hypothesis that might explain this periodic or oscillatory dynamic might be overwintering. An annual increase in *D. suzukii* relative abundance over the seven-year period is indicative of *D. suzukii* overwintering in Maine. There is evidence that while *D. suzukii* can over winter in north temperate US climate zones, survival rates are low [27,28,29]. Experimental studies of ours suggest that the survival rate in Maine is also low [29]. Some winters were more severe for overwintering than others during our seven-year study. The winter of 2017–2018 was one of the colder winters in recent years with extreme temperatures below −26 °C in southern Maine. This was 9–10 °C colder than in 2016 or 2017. Therefore, the large drop in relative abundance from 2017 to 2018 could be due to this extreme harsh winter. A negative correlation between summer relative abundance of *D. suzukii* and the previous winter temperatures is expected. Our analysis of relative abundance from one year to the next showed that mild winter temperatures resulted in higher relative abundance in wild blueberry fields the subsequent year and harsh winter temperatures resulted in lower relative abundance the following year. A similar finding by Briem et al. [16] provided evidence in a seven-year study in Germany that mild winters can positively influence relative abundance of flies. 

Crop production system does not appear to affect relative abundance of *D. suzukii* adults in wild blueberry fields, other than indirectly by determining the number of insecticides that are applied during the period of *D. suzukii* buildup. One would think that because insecticide applications reduce fruit infestation, that this would translate into lower adult *D. suzukii* relative abundance. However, the explosive logarithmic increase in trap captures prior to harvest suggests that insecticides are functioning more as crop protectants from fruit infestation then reducing buildup of adult populations. 

Crop stage does affect relative abundance of flies captured. Fruit bearing fields were found to have significantly higher relative abundance than vegetative fields (no commercially produced fruit present). We have previously shown that wild non-blueberry fruits are important in the buildup and colonizing levels of *D. suzukii* in fruit-bearing fields [38]. However, the existence of *D. suzukii* in vegetative fields also suggests that wild non-crop fruits serve as host reservoirs that can maintain populations in a field when there is no crop until the following year when a susceptible crop will be present. These vegetative field populations may also be important if flies migrate seasonally to crop field habitats. We do not know if this occurs in Maine wild blueberry, but Tait et al. [33] demonstrated that seasonal migration over large spatial scales (ca. 9 km) in Italy facilitates exploitation of resources in patchy environments. Briem et al. [16] also found that different habitats affected relative abundance of *D. suzukii* in Germany, but in their case, forest edges and hedgerows had higher population relative abundances than crop fields. This is opposite of what we found, there is no corollary of alternating fruit crops from one year to the next in other fruit crop systems that we know of, with the exception of strawberries. However, we did not find a discussion of this phenomenon with strawberry production in the literature. 

We also found that the first detection of *D. suzukii* adults in traps was earlier with each year of the invasion. We have not found this phenomenon reported by other researchers. We speculate this may reach a more stable date once the population becomes more adapted to Maine climates, as has been documented with other invasive organisms [17]. However, this also might not be the case. Earlier dates of infestation might reflect climate change. Hotter summers have occurred in Maine over the past decade. In our analysis of summer temperatures in Maine based upon National Weather Service data [49], we found that between 1970 and 1980 there were 449 days that were 26.7 °C or hotter, whereas, between 2008 and 2018 there were 501 days that were 26.7 °C or hotter. This earlier detection of flies in traps was not directly associated with earlier infestation rates in wild blueberry fields. However, we did find an indirect relationship. Earlier detection appears to result in higher relative abundance of flies, and higher relative abundance of flies results in higher fruit infestation rates. 

Fruit infestation rates are relatively low in Maine wild blueberry, ranging from 0–5%. Very little has been reported in the literature about fruit infestation levels by *D. suzukii* in various crops, although several studies have documented economic losses. DiGiacomo et al. [50] reported on a survey of Minnesota raspberry growers that suggested losses ranging from 2–100% of planted crop land and a total loss of U.S. $2.36 million. Asplen et al. [23] cite Goodhue et al. [51] and suggest that if not controlled *D. suzukii* can result in an annual loss of $500 million in western U.S. strawberry and raspberry production. Dal Fava et al. [52] published an economic cost/benefit analysis for fruit production in Italy. Their conclusion was that fruit infestation was not a clear measure of crop loss. It was the frequency of insecticide applications that made profit less likely, especially under high fly pressure. However, it is not the absolute crop loss that many growers worry about, but more the detection of any larvae in fruit that is bound for markets with low tolerance, such as foreign export markets [52]. As mentioned previously, we did find that fruit infestation level is related to adult relative abundance, but additional variation in fruit infestation is explained by year to year variation (we suspect this is due to variation in weather conditions), and insecticide applications. Crop management system was indirectly related to fruit infestation level because management system determines the average number of insecticide applications applied during the late summer. There has not been a lot of data documenting commercial “field-level” effectiveness of multiple insecticide applications in growers commercial fields in reducing fruit infestation. But more specifically, the relationship between increased frequency of applications and the subsequent increased reduction of infestation. A few studies in other states and crop systems have shown that insecticide applications in general reduce fruit infestation [53,54,55]. The average insecticide application frequency for control of *D. suzukii* was observed to be the highest in the high input management system, with the average insecticide application being two per season. However, some commercial fields received three and four applications during the summer. Our data suggests that more than two applications targeting *D. suzukii* during the growing season will have no added benefit in reducing fruit infestation.

A very interesting finding of ours is that sex ratio relative to males has been declining over the seven year period (Figure 4B). When male abundance is compared to female abundance over the entire seven-year period, the proportions of male and female flies appear relatively constant as a function of relative abundance (Figure 4A). Therefore, we are not confident that the trend we have observed in declining sex ratio relative to males is real. This is because the logistic regression on sex ratio that we conducted is highly leveraged by the observed sex ratios in 2012. If this drop in the proportion of males is happening, then one hypothesis is that *D. suzukii* populations in Maine are infected with *Wolbachia* spp. This bacterial infection is known to be lethal to males in several insect species, most notably in Drosophila [56]. *Wolbachia* bacteria has been observed in high prevalence levels in *D. suzukii* in Europe, but only moderate infection rates (7–58%) and no cytoplasmic incompatibility have been observed in North American *D. suzukii* populations [57,58]. At this time, we have not sampled and assayed flies in Maine to verify the presence of this bacterial pathogen. 

The invasion of *D. suzukii* has disrupted well-established effective IPM programs throughout the U.S. [54,55,59]. Some fruit production systems in North America increased insecticide applications dramatically to minimize fruit infestation [55,60]. Biological control and removal of non-crop fruit hosts in the vicinity of crops have been presented as alternatives or complimentary management tactics to insecticides for this invasive pest [19,20,21,22,34,35,39,58]. We have found from two studies in wild blueberry, this one and Ballman and Drummond [39], that increased abundance of non-crop fruit along field edges results in increased fly pressure in the crop as measured by trap capture. We have also found that predation of *D. suzukii* pupae is high in wild blueberry fields [21], but before this study, we did not have any measure of the impact of natural enemies on relative abundance of flies in fields. We showed that natural enemies do reduce fly pressure, but only in fields that have high non-crop wild fruit abundance. We interpret this in the following way. Fields that have high non-crop wild fruit abundance have higher fly pressure resulting in higher rates of reproduction in fruit bearing fields or in non-crop wild fruit refugia along wild blueberry field edges. Under high *D. suzukii* relative abundance and reproduction, natural enemies numerically respond, and under this situation, reduce the higher level pest populations. However, because the fly pressure is so high under this scenario, the amount of reduction in the fly population does not appear to be capable, on its own, of reducing pest pressure to a level that reduces risk for growers. This dynamic needs to be researched further and in other crop systems. We are not suggesting that biological control is of little benefit. Lee et al. [58] have reviewed all of the biological control studies conducted on *D. suzukii* worldwide. They make a strong case that biological control is promising and that it may be influential in lowering risk to growers of crops that are currently vulnerable to *D. suzukii* attack.

*Drosophila suzukii* invasions have resulted in increased insecticide applications to many crops vulnerable to this pest in both North America and Europe. Dal Fava et al. [52] show in an economic analysis of *D. suzukii* management, that it is the increased frequency in insecticide applications than actual damage from fruit infestation that is the serious economic consequence to farmers. This may well be the case for Maine wild blueberry growers as well. We have shown that insecticide applications do decrease the infestation levels of fruit in Maine wild blueberry production, but more applications than two per season appear to reduce the benefit of this management tactic and may only reduce net profit margins for farmers.

We were able to develop action thresholds between 2012 and 2017. A risk-aversion gradient approach was taken [61,62]. A risk-based action threshold is appropriate given the different production systems that exist in Maine for wild blueberry and given that markets range from local fresh market roadside sales, to the export of frozen berries to foreign markets. Therefore, growers can select the threshold level that best suits their capital investment and perceived risk due to *D. suzukii* damage potential. Other than, the first occurrence of flies in traps, we do not know of any other fruit crop IPM programs that utilize action thresholds for management of *D. suzukii*. Wild blueberry growers are currently using these thresholds [2]. The thresholds were well supported by the 2016 and 2017 validation studies. There was only a minimal departure between observed and expected proportion of infested fields resulting from the selection of each specific threshold value. In 2018, while the early harvest tactic was still highly effective, the thresholds did not prove to match up the expected with the observed proportion of fields with infested fruit. So, in two of three years the thresholds worked well. This threshold system does have a buffer for preventing unexpected losses. We recommend that growers take a fruit sample to determine the percent fruit infestation before they reach threshold levels. In this way growers can have confidence in this decision making tool and also catch any damage before it gets too high if the thresholds break down. However, in all three years, the infestation rates in 24 of 35 fields (68.6%) in our validation test were well predicted by our probability model. Even in 2018, most of the infestation levels were between 1-3% and so infestation rates were still relatively low in the absence of insecticides. Clearly, we need to conduct several more years of validation in order to verify that these thresholds minimize the risk for growers, or at the very least, determine how likely the thresholds are to provide accurate predictions. We can only speculate why the thresholds did not perform well in 2018. That summer was one of the hottest summers on record for Maine, with extreme high air temperatures of 35º C for both July and August. We hypothesize that the fermentation bait we recommend either lost its attraction to flies, or that under extremely hot conditions, flies are reluctant to enter the traps. In either case, the threshold based upon trap capture would be underestimating cumulative male fly relative abundance. Briem et al. [16] showed evidence of high temperatures diminishing trap efficacy for *D. suzukii*. Rogers et al. [63] demonstrated that in high tunnel production of fall red raspberries air temperatures exceeded optimal temperatures for development, mating, and oviposition. In addition, although not mentioned by the authors, these temperatures could have been important in their observed low trap captures compared to trap captures within the raspberry crop outside of the tunnels. High temperature regimes might be a phenomenon that needs more attention in the future, especially as climate change is predicted to result in hotter future summers in many north temperate fruit growing regions. 

## 5. Conclusions

*Drosophila suzukii* is a serious pest in Maine wild blueberry characterized by rapid increase during the growing season. It is still in high fluctuating abundance seven years after initial invasion in 2012 and adults are detected in wild blueberry fields earlier in the growing season each year. Mild winters and earlier subsequent detection in the summer, the higher the fly pressure (cumulative trap capture at standardized growing season 705 DD) during the growing season. The level of fruit infestation is determined by year, fly pressure, and insecticide application frequency. Insecticide frequency is determined by the production system. Non-crop wild fruit and predation influence fly pressure in fields. Increase in abundance of non-crop wild fruit along field edges results in increased fly relative abundance. Increased predation in response to higher fly pressure reduces the rate of fly pressure increase. However, predation only appears to operate significantly at high relative abundance of flies, or when high levels of wild fruit are present along field edges and this may not be enough to avert risk of infestation for growers. Early harvest is a highly effective means of avoiding fruit infestation, but there is a cost due to harvest of immature fruit. This cost diminishes as the season progresses, but the likelihood of infestation increases. Action thresholds that were developed and used by growers in 35 commercial fields, based upon cumulative male captures, worked remarkably well in two of three years. Future research will be focused on continued evaluation and possible fine-tuning of the risk-based thresholds.

## Figures and Tables

**Figure 1 insects-10-00205-f001:**
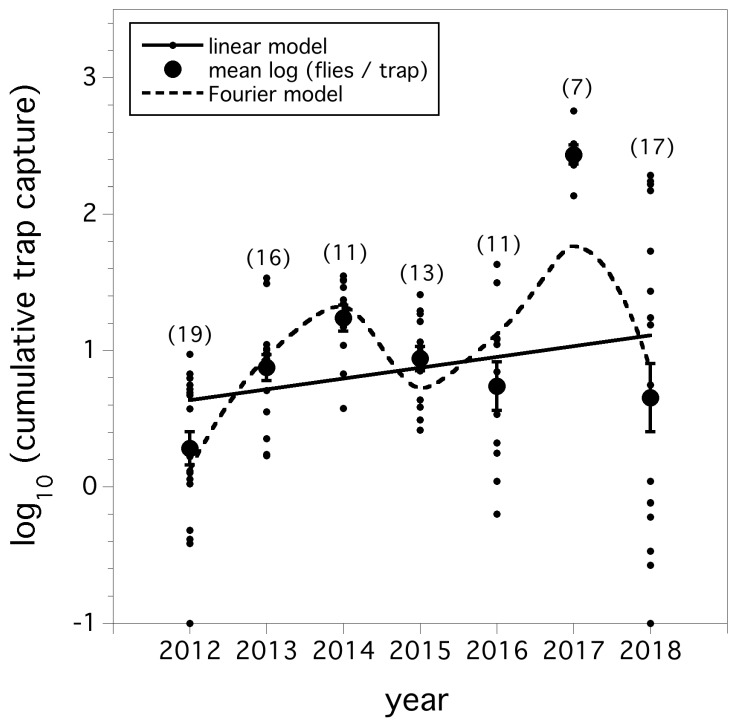
Cumulative trap captures (logarithm transformed) of *D. suzukii* at a standardized growing season of 705 DD (base 10 °C). Small filled circles are individual field captures and large filled circles are annual means (error bars are standard errors). Parentheses above each year are the number of fields where degree-day information was available (total *n* = 97). Solid line is linear model and dashed line is Fourier periodic model.

**Figure 2 insects-10-00205-f002:**
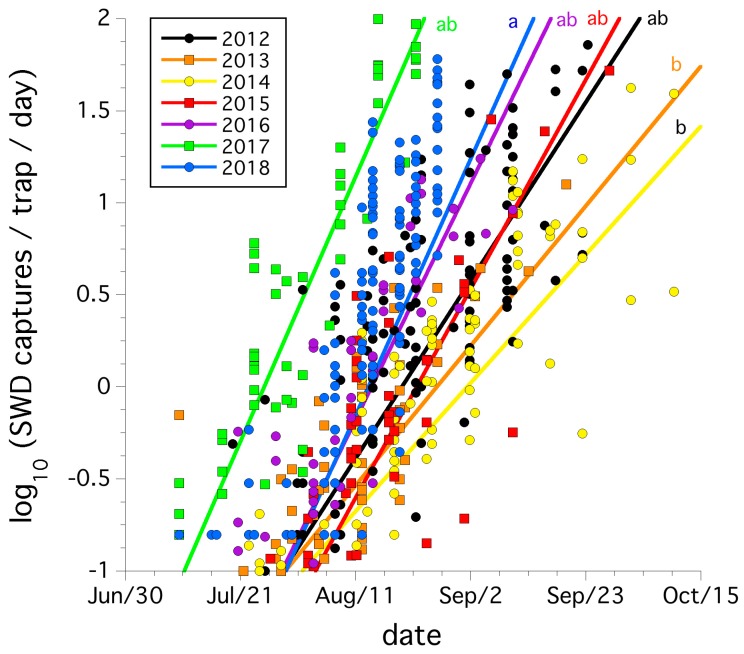
Average trap catches of *D. suzukii* adults per day (logarithm transformed) over each growing season for each field (2012–2018). The same letters next to each regression line signify that slopes are not significantly different as determined by overlap of slope with 95% confidence intervals.

**Figure 3 insects-10-00205-f003:**
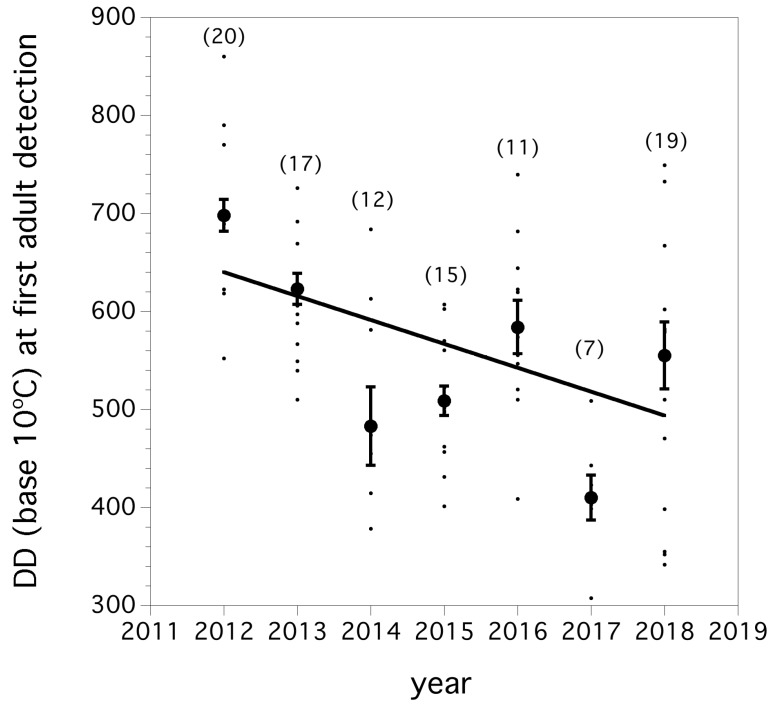
Degree day (base 10 °C) at first fly detection in a trap. Small filled circles are individual field captures and large filled circles are annual means (error bars are standard errors). Parentheses above each year are the number of fields where degree-day information was available (*n* = 101). Solid line is linear regression fit.

**Figure 4 insects-10-00205-f004:**
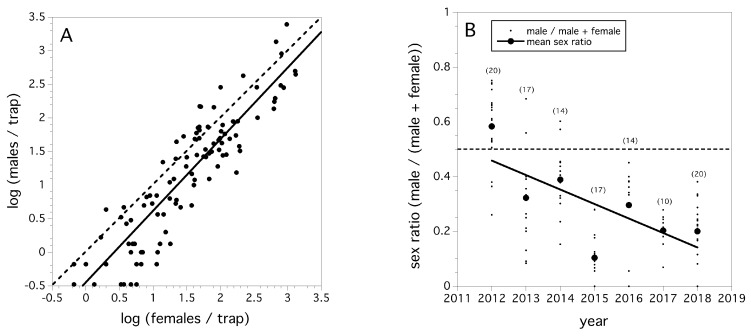
Relationship between male and female trap capture, all fields and all years, solid line is least squares regression fit and dashed line is 50% sex ratio estimate (**A**), and the sex ratio (male: male + female) of total flies captured in each field season by year, small filled circles are individual field sex ratios and large filled circles are mean sex ratios (error bars are standard errors) (**B**).

**Figure 5 insects-10-00205-f005:**
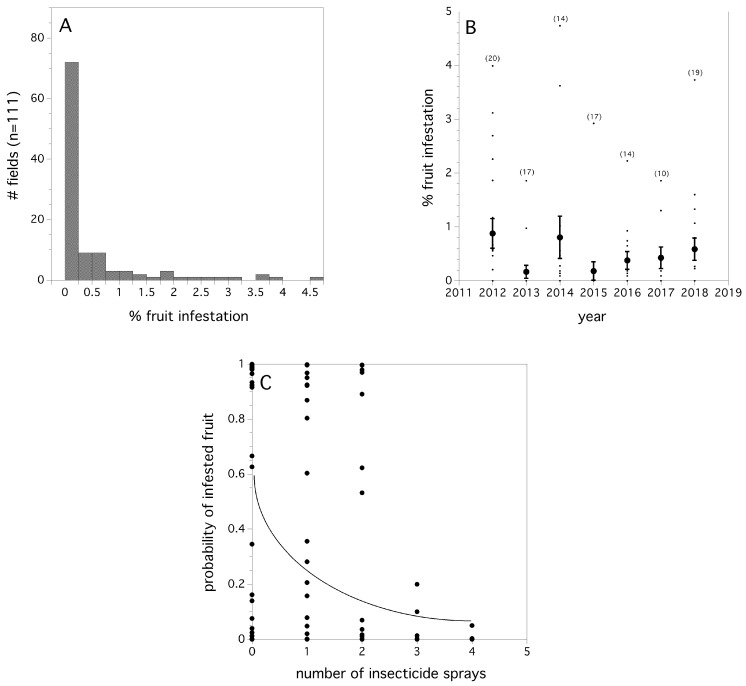
Empirical frequency distribution of fruit infestation levels (%) per field over all years (**A**); the percent fruit infestation in individual fields by year, small filled circles are individual field infestation rates and large filled circles are annual means (error bars are standard errors) (**B**); and the relationship between model estimated probabilities of infested blueberry fruit in a field and the number of insecticide applications applied. Solid line is predicted logistic regression fit (**C**).

**Figure 6 insects-10-00205-f006:**
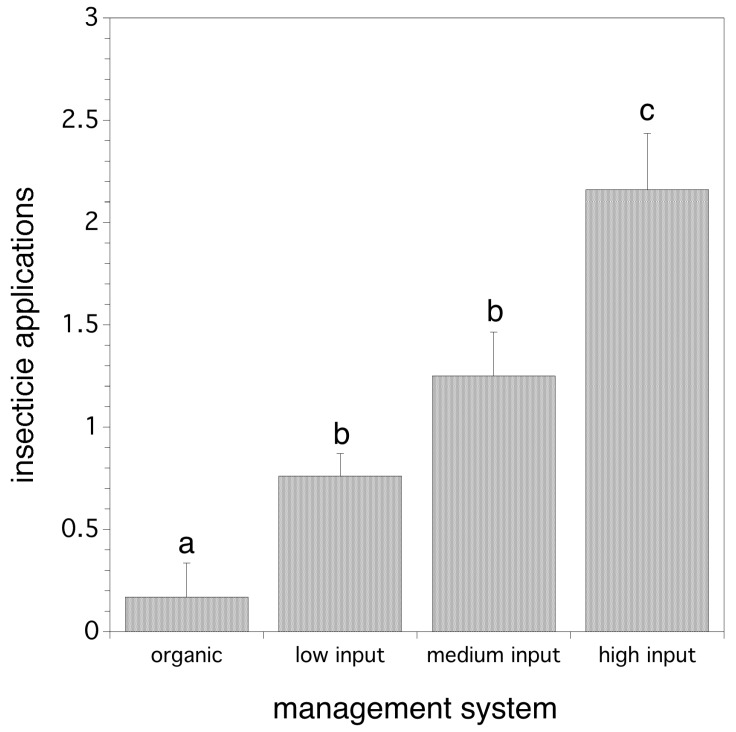
The effect of production system on the frequency (number) of insecticides used during the summer among the four management systems. Error bars are standard errors and the same letters associated with bars by treatment indicate that means are not significantly different from one another (*p* > 0.05).

**Figure 7 insects-10-00205-f007:**
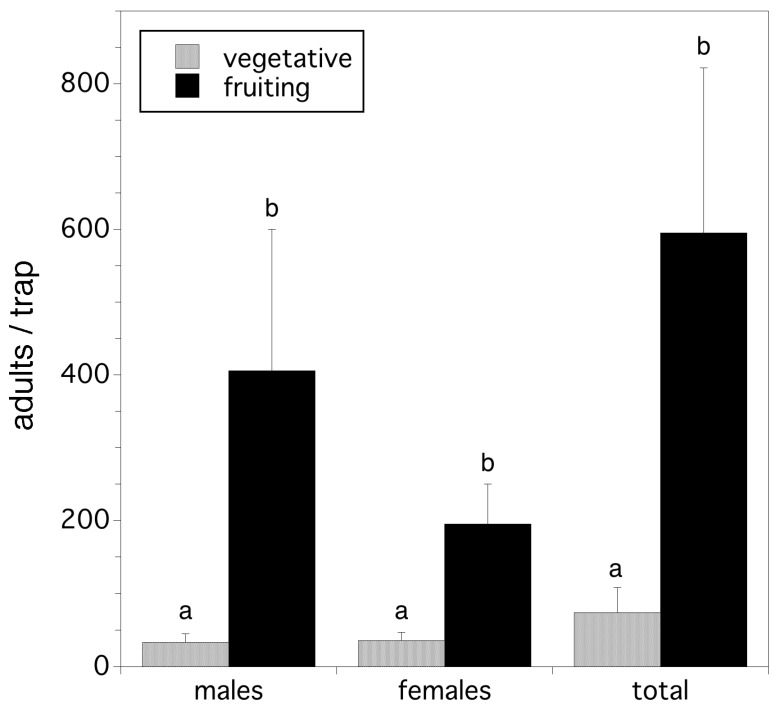
Adult trap captures in vegetative compared to fruiting fields. Error bars are standard errors and the same letters associated with bars by treatment indicate that means are not significantly different from one another (*p* > 0.05).

**Figure 8 insects-10-00205-f008:**
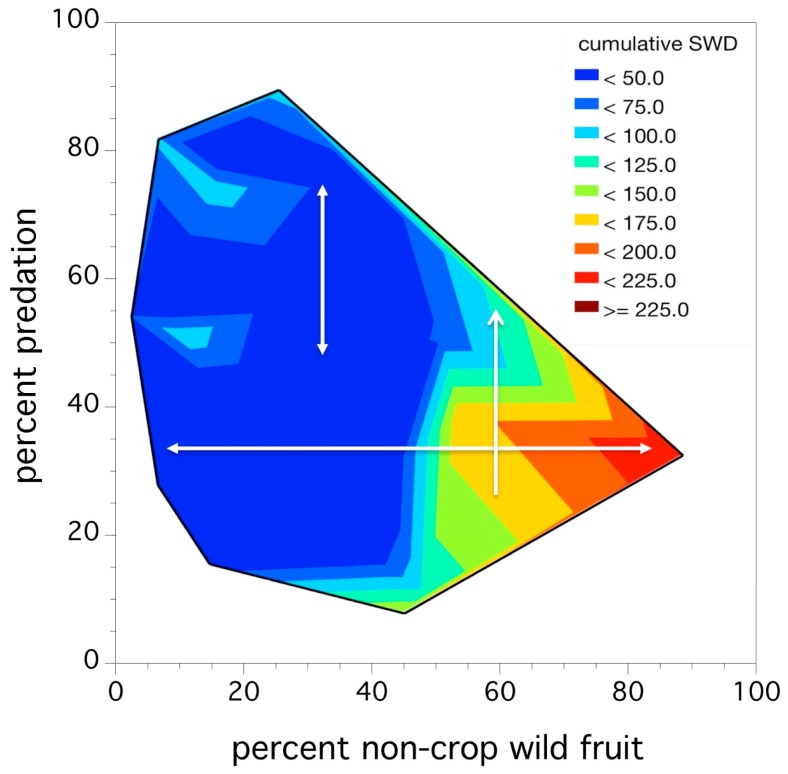
Thermograph of cumulative seasonal relative abundance of *D. suzukii* captures mean flies/trap in fruit-bearing fields as a function of pupal predation (x-axis) and non-crop wild fruit hosts (y-axis). Colors denote *D. suzukii* relative abundance, darker blue regions are low and red regions are high adult relative abundances. White arrows show the change in adult relative abundance as percent predation and percent non-crop wild fruit changes.

**Figure 9 insects-10-00205-f009:**
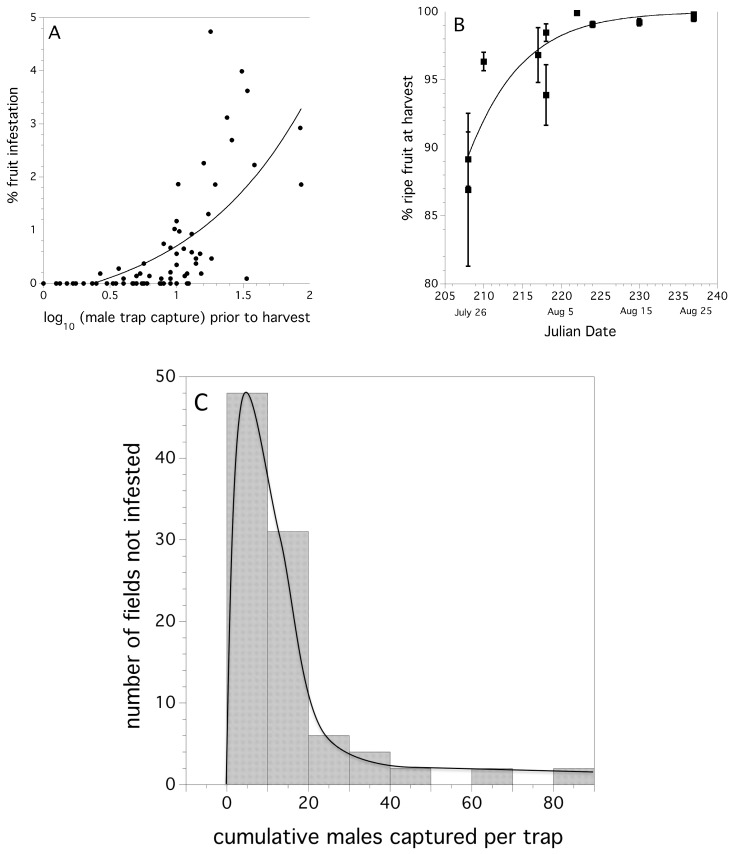
Exponential relationship between percent infested wild blueberry fruit and cumulative male *D. suzukii* trap capture (logarithmic transformed) prior to harvest (**A**), percentage ripe fruit as the harvest season progresses (2016), error bars are standard errors (**B**), histogram for all 92 wild blueberry fields that prior or at harvest were not infested (2012–2017), based on the mean number of cumulative male trap captures in the week before the harvest. The Gamma distribution that is fitted to the data with maximum likelihood methods is also shown in this graph (**C**).

**Table 1 insects-10-00205-t001:** Sampling frequency (number traps per field and trapping interval) for *D. suzukii* annual monitoring of adult relative abundance.

Year	Number of Fields ^a^	Traps per Site	Starting Date ^c^	Sampling Interval (Days)
2012	20 (19)	4–5	8 or 30 June	7
2013	17 (16)	2	3–5 July	7
2014	14 (11)	3 ^b^	8–9 July	7
2015	17 (13)	4	16–24 July	3–7
2016	14 (11)	3	15–21 July	5–7
2017	10 (7)	3	30 June	5–7
2018	20 (17)	3	2–6 July	3–7

^a^ number of fields (total *n* = 112), in parentheses are the number of fields that a standardized growing season degree day value of 705 could be calculated. ^b^ 7 traps deployed in one Jonesboro, ME, field. ^c^ Starting date refers to when traps were first deployed in fields. In all cases they were deployed prior to the occurrence of adult *D. suzukii* in wild blueberry fields.

**Table 2 insects-10-00205-t002:** Typical ^a^ inputs associated with the four wild blueberry production management systems ^b^ [8,9].

Production Factors	Organic	Low Input	Medium Input	High Input
Pruning	Burned	Burned	Mowed	Mowed
Land leveling	Not land leveled ^c^	Not land leveled	Land leveled	Land leveled
pH management	pH managed	No pH management	pH managed	pH managed
Fertility	No fertilizer	No fertilizer	Reduced Fertilization (every other cycle)	Intensive Fertilization (fertilized every cycle)
Pest, disease, and weed control	Cutting woody weeds, no insecticides, no fungicides	Herbicide, blueberry maggot, mummy berry control with standard pesticides	Scouting, standard and reduced risk pesticides	Scouting, reduced risk pesticides
Treatment of bare spots	Mulch	No mulch	No mulch	Mulch
Irrigation	No irrigation	No irrigation	No irrigation	Irrigation
Pollination (honey bees)	0–2.5 hives/ha	0–5 hives/ha	5–10 hives/ha	15–25 hives/ha
Harvest method	Hand raked	Hand raked	Mechanical Harvest	Mechanical Harvest

^a^ There is variation in grower production methods in each of the production systems. ^b^ Production systems are not mutually exclusive. Some growers manage more than one production system. ^c^ Land leveling involves rock removal and lifting the blueberry sod and grading the underlying soil.

**Table 3 insects-10-00205-t003:** Predictive model for determining cumulative relative abundance ^1^ of *D. suzukii* per Table 1. by the end of 2018 growing season.

Terms in Model	Estimate ^2^	S.E. ^3^	T−Ratio	Probability ^4^
Intercept	11.688	32.669	0.36	0.725
% Wild fruit	1.359	0.565	2.40	0.029
% Pupal predation	−0.024	0.430	−0.06	0.956
% Wild fruit X % pupal predation	−0.063	0.018	−2.22	0.042

^1^ cumulative fly trap capture per trap by end of season, not transformed; ^2^ estimate of model coefficient; ^3^ standard error, precision of the coefficient; ^4^ probabilities in bold denote a coefficient significantly different than 0.

**Table 4 insects-10-00205-t004:** Results of early harvest as a tactic for averting risk of fruit infestation by *D. suzukii*.

Year(s)	Number of Fields	Dates of Harvest	Wild Blueberry Infestation Rate
2012–2015	7	21 July–28 July	0.0
2016	3	26 July–4 Aug.	0.0
2017	1	18 July	0.0
2018	4	30 July–8 Aug.	0.0

**Table 5 insects-10-00205-t005:** Action thresholds for cumulative male *D. suzukii* per trap and the expected probability of infestation the week following last trap capture (1- probability estimated from the two-parameter Gamma density function).

Action Threshold: Cumulative Males Captured/Trap	Probability of Infested Fruit: the Week following Trap Capture
0.25	0.001
0.50	0.005
1.0	0.01
2.0	0.05
3.5	0.1
7.0	0.25
16.0	0.5

**Table 6 insects-10-00205-t006:** Validation of action thresholds in commercial wild blueberry fields (2016–2018).

Year	Action Threshold Cumulative Males per Trap	Number of Fields	Observed Proportion Infestation Rate ^1^ (Proportion Fields ^2^)	Model Predicted ^3^ Proportion Fields Infested
2016	1	3	0, 0, 0 (0)	0.01
	3	3	0, 0, 0 (0)	0.08
	9	5	0, 0, 0, 0.003, 0.005 (0.40)	0.30
2017	1	3	0, 0, 0 (0)	0.02
	3	3	0, 0, 0 (0)	0.08
	6	4	0, 0, 0, 0.006 (0.25)	0.23
	1	3	0, 0, 0.01 (0.33)	0.01 ^4^
2018	3.5	6	0, 0, 0, 0.02, 0.03, 0.05 (0.50)	0.10
	7	2	0, 0.04 (0.50)	0.25
	10	3	0.02, 0.01, 0.003 (1.00)	0.50

^1^ Percent infested fruit sampled within a field. ^2^ Sampled proportion of fields that were infested with *D. suzukii* larvae the week following the action threshold was reached or the field was harvested. ^3^ Expected proportion of fields (predicted from probability model) associated with the selected action threshold that will be infested with *D. suzukii* larvae. ^4^ Bold expectations differ in a large degree from the observed.

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
