# Peer review of "Population Dynamics of Spotted Wing Drosophila (*Drosophila suzukii* (Matsumura)) in Maine Wild Blueberry (*Vaccinium angustifolium* Aiton)"

_insects, 2019, doi:10.3390/insects10070205_

Round 1

Reviewer 1 Report

see attach

Author Response

We thank all three reviewers for their thorough reading of this manuscript and constructive criticisms that have made this manuscript far superior than when it was first submitted. We also apologize for many of our typos and careless mistakes or omissions.

Reviewer #1 suggestions and comments

For this manuscript I suggest “major revision. 

Some of the details of the statistics are in the results section (where they do not belong). 

Moreover, there are a lot of things unclear with respect to the statistical part of this paper. 

1. L 114: I do not know what FAD is.FAD refers to the author Frank Drummond. I have changed the text to read: "derived by one of the authors, F. Drummond". 

2. Table 1 and lines 128-130: if one computes a value of degree days, one has to define the day the accumulation of degree days started, the base temperature and the interval of time that is considered in the calculation. With hourly data for temperature one can get a different number of degree days then with daily temperatures. So please, clarify the computation of degree days. The reviewer is absolutely correct. We left out these details and now we have added the requested information. Table 1 has no reference to degree days, but our column "starting date is confusing" it does not refer to the days that we started accumulating degree days, but refers to the dates that we started trapping spotted wing drosophila adults, in all cases these dates were prior to first detection. Therefore, we have defined starting date in the Table 1 footnote. In line 133 - 134 we clarify the calculation of degree days. We did state that the total degree day accumulation that represents the growing season (1 April to mid-August) was 705 (base 10ºC). BUT, we clarify that maximum and minimum daily air temperatures were used from the nearest weather station to a field and that the calculation of cumulative degree days was initiated on April 1 of each year. 

Line 125 (in the table) is really unclear and the term biofix is used but I do not know 

what it means. Line 125 (now line 125-126) is the Table 1 title if we are not mistaken. We have rewritten the Table title to make it more clear to the reader. The title now reads: "Sampling frequency (number traps per field and trapping interval) for D. suzukii annual monitoring of adult relative abundance."We also defined biofix in what is now Lines 146 - 150 as: " This biofix as described above is the cumulative number of degree days typical in most years that covers that start of plant development in the spring (1 April) to harvest (mid-August). The rationale for selecting a standardized physiological time or biofix is so that relative abundance, based upon trap capture, could be compared from one year to the next."   

L267-274; it is totally unclear what the model is that is fitted to the data. 

Figure 1; I do not know what a “fourth order quadratic model” is; that description makes no sense. I think you want to say that you used a fourth order polynomial to describe the data. That is non-sensical too, because essentially you have 7 years for this function and in a fourth order polynomial you estimate 5 parameters on those 7 years. I know for sure that a sixth order polynomial exactly goes through the means of all 7 years (that can be mathematically proven). We believe that the reviewer misread our intentions for fitting a 4th order quadratic to the time series data. This was most likely because we were not clear. We have described in the text the purpose of our intent as follows. First we fit a linear model to all of the data from 2012-2018 (n=97 fields). We showed that while significant the r2is low and that a linear model might not be the best as the residuals do not suggest a linear model. However, there are probably many factors that affect the trend in SWD relative abundance over the seven years. Our intent in using a 4th order quadratic was NOT to suggest that this is a predictive model or that a quadratic function of years has any specific meaning, but instead we wanted to show that the overall trend in relative abundance may be more non-linear and oscillatory in nature, so we fit a 4th order term, not to the 7 means, but to all 97 fields. The residuals are better behaved about the estimated line and the r2 is higher suggesting that some form of oscillatory dynamics might be occuring, BUT NOT TO SUGGEST THAT A 4th ORDER POLYNOMIAL HAS ANY SPECIFIC MEANING. We agree with the author that as a model a 4th order polynomial has no biological meaning. We also tried a cubic spline, the result was almost identical in suggesting oscillatory dynamics, but cubic splines are even more difficult to conceptualize and for our purposes, the form of the model is not important. We hope that this explains our intent to the reviewer. We have attempted to explain this in the text without being overly verbose. The following is how we have no phrased the paragraph, hopefully this makes more sense: " The logarithm of the cumulative trap capture of D. suzukiiflies at a biofix of degree-day 705 (harvest period) over the seven year period (F(1,95)= 4.602, P= 0.035, r2= 0.046) is shown in Figure 1. Because the Studentized residuals from the linear regression were not characteristic of a linear increase in D. suzukii over time. A linear model is probably too simplistic, but does provide an approximation of a positive trend in logarithm relative abundance changes over time. In order to determine if the trend in logarithm relative abundance might be more likely to be non-linear or oscillatory, we also evaluated a fourth order polynomial model with the hypothesis that between 2012 and 2018, D. suzukii has been increasing, but in an oscillatory manner. This model, also shown in Figure 1, explains more of the variance in logarithm cumulative trap captures (F(4,92)= 8.845, P< 0.0001, r2= 0.271), and the residuals suggest a better choice of hypothesis. However, our intention is not to suggest a fourth order polynomial is a superior model or even that it has specific meaning. We only used the polynomial model to suggest that the trend in logarithm relative abundance might be more complex than a simple linear function of season." 

Legend of fig 2: I think that the vertical axis represents logarithmic transformation (base 10) of the cumulative number found up to a certain divided by the total number of emptied traps up to that same date. Thank you, we have better defined the vertical axis in Fig 2. as: "Log10(SWD trap captures / trap / day)" and we added clarification in the figure caption: " Average trap catches of D. suzukii adults per day (logarithm transformed) over each growing season for each field (2012-2018). Traps were collected and new traps were deployed every sample period."  

Line 316 Sex ratio (square root transformed). Is the sex ratio defines as males/ (males 

+females), males/females, and I do not see the use of square root transforming it. I would suggest to use a binomial model with the number of males the successes and the total number of draws (the number of males plus the number of females).We have adopted the reviewer's suggestion and reanalyzed the sex ratio measures as a binomial variate and used logisitic regression with year as the independent variable. We have included the details of the logistic regression analysis of sex ratio in the methods and also we have reported the results of this analysis in the results. While the logistic regression approach does not change the outcome of our original analysis, we thank the reviewer for the insight into performing this analysis as a binomial event. Because of the new analysis we have replotted figure 5B by using male / female sex ratio and NOT the square root of the male / female ratio. 

            We have rewritten the methods as: "Sex ratio of fly trap captures (males / (males + females)) was assessed over the seven year study using two methods. First the number of captured males vs female trap capture in each field in each year was assessed to determine if total male captures relative to total female trap captures were constant. This hypothesis was tested using linear regression of log transformed trap captures (log males vs log females). The slope was evaluated for significance and if it was different from 1.0. A second analysis was conducted to determine if sex ratio changed over the seven-year period of the study. Sex ratios were analyzed for every sampled field for each year. A binomial logistic regression a model with the number of males the successes and the total number of draws (the number of males plus the number of females) was used to test if sex ratio changed over time. Odds ratios were calculated by year and over the seven-year period to quantify the sex ratio change. 

            We have also rewritten the results as follows: " We found that sex ratio of flies captured in traps was female biased; although, the slope of the regression relating log (female flies / trap) to log (male flies / trap) suggested that the sex ratio (males / (males + females)) is constant across fly population density (F(1,107)= 440.271, P< 0.0001, r2= 0.815, β= +1.065, Fig. 5A). Sex ratio relative to the proportion of males to females declined in a linear fashion between 2012 and 2018 (c2(1)= 10239.6, P< 0.0001, Fig 5B). The odds ratio of sex ratio change per year is estimated at 0.713, and over the entire period between 2012 and 2018 is 0.131."

L343 and further down: what is abundance? It could be the cumulative count, the per trap average of the cumulative count Can you define how you measured abundance?We have defined relative abundance as abundance of flies caught in traps as follows: "We refer to trap capture abundance of flies as relative abundance throughout this study because we realize that trap capture is only an index of absolute abundance as trap captures are affected by behavior and weather conditions" (under section 2.4 in Methods). We have also changed all references to abundance to "relative abundance."  

Does table 3 mean that abundance = 11.688+ 1.359 (% wild fruit) 0.024 (%pupal predation) 0.063 ((% wild fruit) *(%pupal predation))??  The interpretation of the Table 3 coefficients in the linear regression model is: SWD relative abundance = 11.688 + 1.359 * (% wild fruit) - 0.024 (%pupal predation) - 0.063 * ((% wild fruit) *(%pupal predation)).   

Figure 7 is “cumulative SWD” summed over the full season and over all traps?? 

Figure 7 represents the data as a heat map described in Table 3. The actual cumulative fly trap capture is the mean fly capture per trap per field, summed over the duration of the study (July - August 2018). This is now described in the figure caption. The caption now reads: "Cumulative (seasonal relative abundance) of D. suzukiicaptures (mean flies / trap) in fruit-bearing fields as a function of non-crop wild fruit hosts and pupal predation. Darker blue regions are low and red regions are high adult relative abundances. White arrows show the change in adult relative abundance as non-crop wild fruit and predation changes."

L383 is Julian Date the same as calendar day? If yes, please use that term instead of Julia Date. Julian date is a numerical coding of calendar date where January 1 is Julian Date 1. We have defined Julian date in the formula by writting: "% ripe fruit = 100 / [1 + e(30.903 - 0.159 * Julian Date)], P= 0.002, r2= 0.795), where Julian date is a recoded calendar date, where January 1 is day 1." 

Figure 8C it is not clear whether on the horizontal axis you mean cum males per trap over the full season or up to a specific date. We have attempted to clarify the axes labels on the plot of the probability density function. It is now written as: "gamma theoretical distribution and empirical frequency distribution (2012-2017, n=92 fields) comprising wild blueberry fields sampled that are represented by the mean number of cumulative male trap captures just prior to harvest and the likelihood of not being infested the following week after the cumulative trap capture (x-axis) (C)."

Typos: 

L135 I do not know what “degree date” is, I think you want to say the cumulative degree days.Yes, thank you, we fixed this.

L273 where is the decimal point in 0001? We fixed this, thank you.

L372 and L383 you use different multiplication signs ‘x’ and “*”, I think that * is the better one. We replaced all "x" with *

In a lot of places the letters of the species D. suzukii are not in an italicfont like D. suzukii. We have fixed this, thank you.

Bonferroni is with two r’s. We have fixed this, thank you.

“logarithm transformed” should be “logarithmically transformed”. We have fixed this, thank you.

Should “fit” be “fitted”?As far as we know either form can be used and so we have remain using fit.

Reviewer 2 Report

The study presented in this manuscript is very interesting. It refers to a very important pest from the economical point of view and to a long-term study, what is hard to found in the scientific literature regarding pests. The outcomes of authors' experiments can be very valuable for a successful management of Drosophila suzukii. However, I would like to point out some problems that, to my opinion are in the current version of the manuscript.

I found the wording is at some points a little complicated and imprecise. I am not a native English speaker and, therefore, my knowledge of the language is, for sure, not as good as that of the authors. However, for years, I have read many scientific articles and I have found this manuscript particularly difficult to follow. I would advise the authors to try to simplify the sentences and make them as precise as possible.

Generally speaking, I think that authors have to work to make the Mat & Met sections more understandable and to better discuss their results.

Material and Methods. In my opinion, information in this section has to be reorganized. As far as I understood, 2.1 and 2.2 explains the sampling procedure that has been conducted in all fields (12-20). Section 2.3 explains some generalities concerning statistical analysis and modeling, whereas detailed info on different data analysis/modelling are in sub-sections 2.4 to 2.8. I think this is a quite confusing way of organizing the information because puts at the same level the sampling procedures (2.1, 2.2 & 2.3) and data analysis/modelling (2.4 to 2.8). I would suggest to start by explaining the different data analysis/modelling and (2.4 – 2.8) and explain the procedures to obtain the data at the first mention. Subsequent use of the same data should refer to the detailed information about the procedure/ methodology, as it is already done (e.g. L122-123). I am not 100% sure that this way of organizing the info will work, if not authors should explore other ways to explain their Mat & met.

L 59. Add Gabarra et al (2015) BioControl 60: 331-339

L80. Italicize D. suzukii. This mistake is common through the text (L90, 97,115 …). Please, revise.

L87. Change ‘diam.’ to diameter

L95. The ‘Traps were replaced with new traps’ sounds redundant.

L113-123. Authors describe that traps were hung in the crop/forest interface. But, in section 2.5 authors mentioned that traps were hung above the blueberry canopy. So, what are the traps used in section 2.1 used for? Please, clarify.

L118. I understand that for each modeling/ analysis you use a sub-sample of the 12-20 fields that authors mentioned in L80. Is that right? Please clarify.

L 145. Italicize Rhagoletis mendax

Results section. I want to note that I had a limited understanding of some of the analysis done by the authors, especially what refers the modeling because it is out of my field of expertise.

Figures are of very poor quality. Standard errors are barely visible in Fig 1, 3, 4, 5.

In Figures 1, 3, 4B & 5B, delete year 2011 from X-axis. It has not been sampled.

Figure 4 caption is quite confusing. Please, reword. Note that Fig 4B is not described in the caption.

In Figure 6 they are missing in several error bars (e.g. organic and high input in 6A). Please, try to simplify the figure caption.

L306-307. How do authors calculate the probability of blueberry infestation? Which is the model?

L325-327. First part and last part of the sentence say the same! Where is this data shown? Please, reword.

L 332 & 337. Please do not show in the same figure 6 data explained in 2 different subsections. Change 6A to 6 and 6B to 7.

L353-354. Do not capitalize 'only'. The sentence “…relative abundance in the context of the wild….” is difficult to understand

L355. A parenthesis is missing

Discussion.

L 416-424. Most of the information in these lines fit better in the introduction

L436: Any reference suggesting that D. suzukii adults may feed on fungi? Check Fountain et al (2018) Alimentary microbes of winter-form Drosophila suzukii. Insect Molecular Biology and some of the references that these authors use.

L449-451. Why do authors state that the oscillating nature ….. suggest that predation is dampening population increase? Please explain better

L472-475. D. suzukii catches in vegetative fields are very low, so It is difficult to understand that they serve as reservoirs, especially in areas with fruiting fields and suitable wild hosts. Explain better what is the contribution of the vegetative fields to the maintenance of the population in the area.  

L485-487. It is very speculative. Reword.

L488-494. I found this paragraph interesting but the wording quite confusing. I would suggest to rephrase it. I have an additional comment. According to Del Fava et al., not the damage itself but the increased use of insecticides is what reduces economical profit in crops susceptible to D. suzukii. But authors state in L 325-326 that management system and, therefore, the number of insecticides (Fig 6A) does not determine the pest abundance. Thus, is high usage of insecticides in the non-organic managed fields  useless? I think authors should discuss this issue in this paragraph.   I think authors should discuss this issue in this paragraph

L511-512. What do authors mean with “ we are not confident……is real.”?

L512-517. Explain better how you relate your results to Wolbachia.

L530-534. Rephrase and simplify.

L536. Reference 57 is missing in the References list

Author Response

We thank all three reviewers for their thorough reading of this manuscript and constructive criticisms that have made this manuscript far superior than when it was first submitted. We also apologize for many of our typos and careless mistakes or omissions.

Reviewer #2 suggestions and comments

Comments and Suggestions for Authors

The study presented in this manuscript is very interesting. It refers to a very important pest from the economical point of view and to a long-term study, what is hard to found in the scientific literature regarding pests. The outcomes of authors' experiments can be very valuable for a successful management of Drosophila suzukii. However, I would like to point out some problems that, to my opinion are in the current version of the manuscript.

I found the wording is at some points a little complicated and imprecise. I am not a native English speaker and, therefore, my knowledge of the language is, for sure, not as good as that of the authors. However, for years, I have read many scientific articles and I have found this manuscript particularly difficult to follow. I would advise the authors to try to simplify the sentences and make them as precise as possible.

Generally speaking, I think that authors have to work to make the Mat & Met sections more understandable and to better discuss their results. 

We thank the reviewer for the insight on the organization of the manuscript. We have attempted to make the manuscript more clear by rereading the manuscript carefully and rewriting the text where we felt it might have been confusing.

Material and Methods. In my opinion, information in this section has to be reorganized. As far as I understood, 2.1 and 2.2 explains the sampling procedure that has been conducted in all fields (12-20). Section 2.3 explains some generalities concerning statistical analysis and modeling, whereas detailed info on different data analysis/modelling are in sub-sections 2.4 to 2.8. I think this is a quite confusing way of organizing the information because puts at the same level the sampling procedures (2.1, 2.2 & 2.3) and data analysis/modelling (2.4 to 2.8). I would suggest to start by explaining the different data analysis/modelling and (2.4 – 2.8) and explain the procedures to obtain the data at the first mention. Subsequent use of the same data should refer to the detailed information about the procedure/ methodology, as it is already done (e.g. L122-123). I am not 100% sure that this way of organizing the info will work, if not authors should explore other ways to explain their Mat & met.

We have read over the reviewer's comments about the organization and considered carefully the comments put forth to improve the manuscript. We decided to keep the organization the same because we felt that first describing the data collection used in the analyses, ie. the sampling of  adult relative abundance and larval fruit infestation was central to all of the analyses. However, we have taken the reviewer's comments to heart and we have tried to make the subsections 2.4-2.8 more clear by stating the purpose and hypothesis(es) at the very beginning of each subsection.  

L 59. Add Gabarra et al (2015) BioControl 60: 331-339. Thank you for the reference. It is very comprehensive. We have added it as suggested.

L80. Italicize D. suzukii. This mistake is common through the text (L90, 97,115 …). Please, revise. We thank the reviewer for pointing this out and we have now italicized all instances of D. suzukiinow.

L87. Change ‘diam.’ to diameter.We have made this change.

L95. The ‘Traps were replaced with new traps’ sounds redundant.We decided to keep this sentence in the methods because it may not be a standard practice to replace traps with "new" traps at every sample date.

L113-123. Authors describe that traps were hung in the crop/forest interface. But, in section 2.5 authors mentioned that traps were hung above the blueberry canopy. So, what are the traps used in section 2.1 used for? Please, clarify. We clarified our trap placement protocol by adding: "The exception to this trap placement protocol was for the crop cycle study, described in subsection 2.6".

L118. I understand that for each modeling/ analysis you use a sub-sample of the 12-20 fields that authors mentioned in L80. Is that right? Please clarify.No, this is not correct. We have not been clear. Table 1 lists the 112 individual fields that we sampled in the entire study. Because each study (subsections 2.4-2.8) uses a different number of the 112 fields due to small amounts of variation in dependent and independent variables that were measured over the entire study period (7 years), not all studies used the entire data set for analysis. We have now written the methods to read: "The number of fields sampled ranged from 10 to 20 fields per year, listed in Table 1. The number of fields used in each study are described in subsections 2.4-2.8 and in the results. Each study had a small variation in the number of total fields analyzed due to differencesin dependent and independent variables that were measured over the entire study period (7 years)."

L 145. Italicize Rhagoletis mendax. Thank you for catching this error. We have now italicized Rhagoletis mendax.

Results section. I want to note that I had a limited understanding of some of the analysis done by the authors, especially what refers the modeling because it is out of my field of expertise.

Figures are of very poor quality. Standard errors are barely visible in Fig 1, 3, 4, 5.We have added high resolution (600 dpi) figures now in the manuscript.

In Figures 1, 3, 4B & 5B, delete year 2011 from X-axis. It has not been sampled.We have kept 2011 in the x-axis because if we delete it, our graphics package will place the 2012 data on top of the y-axis and it makes it very difficult to see the data. We apologize for this, but we have tried to fix this and we can not make the software space the data away from the axis. 

Figure 4 caption is quite confusing. Please, reword. Note that Fig 4B is not described in the caption.Thank you very much for noting that we had mistakenly forgot to provide the (B) for the caption for Figure 4 B. We also have added some descriptors to make the captions more clear. 

In Figure 6 they are missing in several error bars (e.g. organic and high input in 6A). Please, try to simplify the figure caption. We have enlarged the higher resolution figures so that the error bars are now visible. Thank you. 

L306-307. How do authors calculate the probability of blueberry infestation? Which is the model? The statistical model is a logistic regression. We have clarified this in the results. We have added the following text in the results after mentioning the logistic regression: "The coefficient for the logistic regression estimation of percent fruit infestation as a function of insecticide application was: -1.281 +0.271." We hope that this important point is now more clear, thank you.

L325-327. First part and last part of the sentence say the same! Where is this data shown? Please, reword.We have now rewritten the sex ratio secti0n of the manuscript and reanalyzed the data as requested by another reviewer.

L 332 & 337. Please do not show in the same figure 6 data explained in 2 different subsections. Change 6A to 6 and 6B to 7. We were trying to save space in the manuscript. But you make a good point. We have now made Figure 6A, Figure 6 and Figure 6B Figure 7.

L353-354. Do not capitalize 'only'. The sentence “…relative abundance in the context of the wild….” is difficult to understand. We have removed the capitalization of "ONLY".

L355. A parenthesis is missing. Thank you. We have added the right parenthesis.

Discussion. 

L 416-424. Most of the information in these lines fit better in the introduction. This is probably true, but we included these two sentences in the beginning of the discussion to provide context because "wild blueberry" is a crop that most researchers are not familiar with. Therefore, we hope that it is permissible for the sake of clarity to keep these two sentences in the discussion.

L436: Any reference suggesting that D. suzukii adults may feed on fungi? Check Fountain et al (2018) Alimentary microbes of winter-form Drosophila suzukii. Insect Molecular Biology and some of the references that these authors use.We actually have observed D. suzukiifeeding on fungi, toadstools, and so we have included this in the discussion.

L449-451. Why do authors state that the oscillating nature ….. suggest that predation is dampening population increase? Please explain better. We have added a few more sentences and referenced Figure 1. We have now written: "Our data also suggests that this increase is more likely to be oscillating, or at least stochastic in a non-linear manner and not a simple linear trajectory. We base this on Figure 1, but we have no insight into the factors that may result in such a non-linear trend." 

L472-475. D. suzukii catches in vegetative fields are very low, so It is difficult to understand that they serve as reservoirs, especially in areas with fruiting fields and suitable wild hosts. Explain better what is the contribution of the vegetative fields to the maintenance of the population in the area. The reviewer is correct, we were not very clear in our thinking. We have now rewritten this section to read: "The D. suzukiipopulations maintained in vegetative fields, despite the absence of a crop are most likely supported by wild fruit hosts. This corroborates findings by Ballman et al. [39] who found high reproduction in wild fruits surrounding wild blueberry fields."    

L485-487. It is very speculative. Reword.We have done so and also stated that this is speculation. The sentences now read as: "We have not found this phenomenon reported by other researchers. We speculate this may reach a more stable date once the population becomes more adapted to Maine climates, as has been documented with other invasive organisms [17]. However, this might not be true, the earlier dates of infestation might be reflecting climate change and the hotter summers that have been occurring in Maine over the past decade. In our analysis of summer temperatures in Maine based upon National Weather Service data [49], we found that between 1970 and 1980 there were 449 days that were 26.7 ºC or hotter, whereas, between 2008 and 2018 there were 501 days that were 26.7 ºC or hotter."

L488-494. I found this paragraph interesting but the wording quite confusing. I would suggest to rephrase it. I have an additional comment. According to Del Fava et al., not the damage itself but the increased use of insecticides is what reduces economical profit in crops susceptible to D. suzukii. But authors state in L 325-326 that management system and, therefore, the number of insecticides (Fig 6A) does not determine the pest abundance. Thus, is high usage of insecticides in the non-organic managed fields  useless? I think authors should discuss this issue in this paragraph.   I think authors should discuss this issue in this paragraph. 

We thank the reviewer for her/his thoughts. We actually do not state that insecticides do not work, in fact our analysis has shown that they do, but that applications of more than two appear to be not effective. We have now cited Del Fava et al. and work the idea of increased insecticide applications not fruit infestation as being the real damage and cost to farmers. Thank you very much for this insight. This is what we have now stated "As stated above D. suzukiiinvasions have resulted in increased insecticide applications to many crops vulnerable to this pest in both North America and Europe. Dal Fava et al. [50] show in an economic analysis of D. suzukiimanagement, that it is the increased frequency in insecticide applications than actual damage from fruit infestation that is the serious economic consequence to farmers. This may well be the case for Maine wild blueberry growers as well. We have shown that insecticide applications do decrease the infestation levels of fruit in Maine wild blueberry production, but more applications than two per season appear to reduce the benefit of this management tactic and will only reduce net profit margins for farmers.     

L511-512. What do authors mean with “ we are not confident……is real.”?To clarify our point we have now written it to read: "Therefore, we are not confident that the trend we have observed in declining sex ratio relative to males is real. This is because the logistic regression is highly leveraged by the observed sex ratios in 2012."

L512-517. Explain better how you relate your results to Wolbachia.We have added a reference and an explanation as follows: "In addition, because we have been observing an increase in relative abundance over the 7-year period, a drop in male sex ratio would not be hypothesized to a cause such as Wolbachiaspp. infection. This bacterial infection is known to be lethal to males in several insect species, most notably in Drosophila [54]. Wolbachiabacteria has been observed in high prevalence levels in Europe, but high infection rates and reproductive manipulation have not been observed in North American D. suzukiipopulations [55]. We have not sampled and assayed flies in Maine to verify this."

L530-534. Rephrase and simplify.We have hopefully clarified the paragraph by selectivelt adding a few key words.

L536. Reference 57 is missing in the References list. References have no been corrected, thank you.

Reviewer 3 Report

This is a valuable study that has practical implications for local management of this important invasive pest in wild blueberry crops in Maine. The authors have done considerably a lot of works over a period of 7 years towards understanding the local fly population dynamics and have developed action thresholds based upon cumulative male captures, which may help growers with control decision.

My major suggestion is to standardize the number of trap capture of adult flies based on per week or per day capture for all data analyses, because the sampling interval varied among fields and years.    

I think all the figures need to be improved for the quality and some of them are not very clear (e.g. Figs. 4,5,8). 

Minor comments:

Line 58-59: The following paper recently reviewed biological control of SWD and could be cited additionally here for more comprehensive information on this issue: Lee J.C., X. Wang, K.M. Daane, K.A. Hoelmer, R. Isaacs, A.A. Sial, V.M. Walton.  2019. Biological control of spotted-wing drosophila – current and pending tactics.  JIPM 10: 13, 1-9. doi: 10.1093/jipm/pmz012). 

Line 80: should be 10-20 fields (based on Table 1)? Same for Line 118 (11-20 fields?) 

Line 130: 750 DD here but 1270 DD on Figure 1. 

Line 135: degree date (day) 

Line 138-139: for the GLM model, you do not need to transfer the data.

Author Response

We thank all three reviewers for their thorough reading of this manuscript and constructive criticisms that have made this manuscript far superior than when it was first submitted. We also apologize for many of our typos and careless mistakes or omissions.

Reviewer #3 suggestions and comments

This is a valuable study that has practical implications for local management of

this important invasive pest in wild blueberry crops in Maine. The authors have

done considerably a lot of works over a period of 7 years towards understanding the local fly population dynamics and have developed action thresholds based upon cumulative male captures, which may help growers with control decision.

My major suggestion is to standardize the number of trap capture of adult flies

based on per week or per day capture for all data analyses, because the sampling interval varied among fields and years.

We want to thank the reviewer for the advice of standardizing trap capture by the number of days trapped between trapping intervals. This is important for Figure 2 and we have standardized the data in this way. All of our other analyses were standardized by a degree day biofix of 705 degree days, except in the 2018 analysis of wild fruit and predation, we used used cumulative fly capture per trap for the season and all fields were trapped for the exact same time interval. 

 I think all the figures need to be improved for the quality and some of them are not very clear (e.g. Figs. 4,5,8). Thank you, for the manuscript submission, we submitted low quality figures. We now have incorporated 600 dpi figures and we have relabeled some of the axes and rewritten many of the figure captions in order to make the figure more clear.

Minor comments:

Line 58-59: The following paper recently reviewed biological control of SWD

and could be cited additionally here for more comprehensive information on

this issue: Lee J.C., X. Wang, K.M. Daane, K.A. Hoelmer, R. Isaacs, A.A. Sial,

V.M. Walton. 2019. Biological control of spotted-wing drosophila – current and

pending tactics. JIPM 10: 13, 1-9. doi: 10.1093/jipm/pmz012).  Thank you, this article had not been available when we submitted this manuscript. We have incorporated it into our discussion.

Line 80: should be 10-20 fields (based on Table 1)? Same for Line 118 (11-20

fields?).Thank you, we have fixed our errors.

Line 130: 750 DD here but 1270 DD on Figure 1.Our mistake, thank you for catching this, we have corrected it.

Line 135: degree date (day).Thank you, we have fixed this.

Line 138-139: for the GLM model, you do not need to transfer the data. The reviewer is correct, we could have used a log-normal error term for the distribution. We did transform the data logarithmically in order to develop a model with reliable P-values. Therefore, we decided to use the transformations. 

Round 2

Reviewer 1 Report

comments are in pdf file

Author Response

Reviewer 1

Response to three points. 

Because the authors do not clarify three things appropriately I advise minor revision.  

      The computation of degree days is still not clear. I see that with  -A minimum temperature of 15 C and a maximum temperature of 25 C the amount added to the total degree days is 10 degree day (((15+25) / 2 ) - base of 10ºC) = 20 DD. -A minimum temperature of 2 C and a maximum temperature of 9 C the amount added to the total degree days is 0 degree day. -But what is the amount added if a minimum temperature of 8 C and a maximum temperature of 25 C occurs? This would be: (25 + 8) / 2 = 16.5 and 16.5 - 10 = 6.5 DD. 

      Thank you for this comment we now have made this more clear to the readers by stating exactly what we did. We now state:  "In order to calculate the number of degree days accumulated for each day, we added the maximum and minimum air temperatures for that day and divided by two to estimate the average temperature for that day ([Max + Min] / 2 = Average Daily Temperature). Then the developmental threshold, in our case, 10ºC was subtracted from the daily average temperature to derive the number of degree days ([Average Daily Temperature - Developmental or Base Temperature] = Degree Days). If the resulting degree day estimate is equal to or greater than 0.0, then this number of degree days is added to the cumulative degree days to this point. If the number of degree days derived for that day is less than 0.0, then 0.0 is added to the cumulative degree days). "   

Figure 1 and 3 and 5; I did not misread your intentions, but I disagree with the way you just pose a fourth order polynomial (you should have tried different models, and then with model selection based on AIC or a likelihood ratio test you could have made a choice between these models; then you would have seen whether the better fit is really better because you weigh it against the extra parameters in the model). I do see in all the three figures mentioned that a linear model is not the best one. However, why the authors choose to plot a fourth order polynomial in fig 1 and not something else in figures 3 and 5 I do not get. I advise them to remove the fourth order polynomial from figure 1 and state in the text for all these figures that the linear model seems not to be the best one. In all three cases the relationships is nonlinear. 

Thank you we did what you suggested and compared several different non-linear models using AICc and conducting a chi-square test on the deviance (-2 log likelihood statistic) of the linear model with each of the non-linear alternative models. The methods now read: 

" First, to determine if logarithm of annual fly captures per trap over the season (using biofix of 705 DD as length of season) increased linearly or nonlinearly over time or were characterized by periodicity (2012-2018), we fit the following models for testing this hypothesis: 1) a linear generalized model using maximum likelihood and year as the independent variable; 2) non-linear generalized model, polynomial in form (fourth order) regression; 3) a penalized piece-wise spline regression (to prevent over-fitting of the data); and 4) a Fourier sine/cosine non-linear model [42]. The piece-wise spline and the Fourier models were fit to the time series data using a mixed model where the spline or Fourier coefficients are treated as random effects estimated as best linear unbiased predictors [42].  Visual inspection of model residuals compared to model predictions and AICc (Akaike information criterion corrected for small sample sizes) were used to provide evidence for selection of a model that would best support either a linear increase or the non-linear alternative models representing periodic or oscillatory dynamics. In addition, we tested if the alternative non-linear models significantly reduced the unexplained variance in the base linear model by comparing the deviances (-2 log likelihood statistics) of the non-linear models to the base linear model with the Chi-square distribution. 

and the Results read: 

" The logarithm of the cumulative trap capture of D. suzukii flies at a biofix of degree-day 705 (harvest period) over the seven year period (F(1,95)= 4.602, P = 0.035, r2= 0.046) is shown in Figure 1. A four parameter Fourier model comprised of sine and cosine coefficients was the best of the alternative models at reducing the deviance (-2 log likelihood statistic) compared to the linear model (P < 0.001). The AICc was 235.4 for the linear model and 193.9 for the Fourier model. Figure 1 shows the predicted model for both the linear and Fourier model. While the model residuals appear visually to be better suited for the Fourier model, a paired T-test showed no evidence (P > 0.05) to suggest significant differences between the two model residuals, although the residuals were larger on average for the linear model. The linear model suggests a steadily increasing fly relative abundance over the seven-year period, while the Fourier model suggests a periodic fluctuating dynamic."         

Original figure 8C (now 9C) 

I think that the authors try to say the following We show a histogram for all 92 wild blueberry fields that at harvestwere not infested (2012-2017), based on the mean number of cumulative male trap captures in the week before the harvest. The gamma distribution that is fitted to the data with maximum likelihood methods is also shown in this graph. 

Thank you our version is not very clear. Yes, we were attempting to state what you more concisely state above. We now have included your text into the figure caption with one minor addition, instead of "at harvest" it reads  "prior to or at harvest".

In a lot of places the letters of the species D. suzukii are not in an italic font like D. suzukii

We have fixed this, thank you. I still see one non-italic around line 90. Thank you again for pointing this out, we have gone through the manuscript again and we believe that we have fixed this.

Reviewer 2 Report

I have gone through the revised version and I realized that authors did not understood the scope of my previous comments and suggestions. When in my first revision I wrote “Generally speaking, I think that authors have to work to make the M&M section more understandable and to better discuss their results”, I meant that authors should attempt a deep reorganization of the information in both sections. I probably did not explain myself well enough. Sorry about that. In the present version, mostly they have add to the text explanations to my queries, but they did not succeed to have a more concise, precise and well structured manuscript. The result is a longer manuscript (120 lines = two pages longer) that, at some points is disorganized and repetitive. I think it is really a pity because these data that really merit publication. However, authors have to critically read the manuscript and make a deep reorganization of the information. Then, this can be a very good paper. Below, there is a non- comprehensive list of my comments and suggestions. I hope they may help the authors to improve the manuscript.

COMMENTS

L45. A parenthesis is missing

L84. Add reference to Table 1. Condense all the information regarding the characterization of the 112 fields here in this paragraph, for example information in L 130-133.

L94, L 128, L267, etc. There is no need to repeat the sampling period unless authors feel that it will be important to precise again this information.

L96. Authors stated that trap placement was uniform for all the study except for that reported in subsection 2.6. However, in section 2.6 there is no particular information about traps deployment. Do authors refer to subsection 2.7? Again, no information about a different placement for traps is described in 2.7. The only information about different deployment of traps is in subsection 2.4 (L133-136). It is quite confusing.

L128-130 and L136-138 can be merged in a single sentence.

L169. Add order and family after Currant

L173-182 How was measured sex ratio? As males / (males+ females) or as males / females? Please, clarify. If both were used, use different terms to refer to these two different concepts.

L223-226 What is different from what was described in subsection 2.1? May be it is not necessary to repeat the information.

L227. Italicize D. suzukii

L236-237. I understood that all results from this subsection were obtained from the sub set of 20 fields (10+10). If so, this was already explained in L222-223 and there is no need to repeat it again.

L266 should read “(see methods in subsection 2.1)

L267-268. Only data of fields that were harvested before to detect Ds males in the traps were used in this analysis? Why? Clarify.

L274. Total number of fields = 112 (Table 1). So, 92 are not all the fields.

L293-294. Is this sentence complete?

L300-303. In my opinion, this is not something to put in Results. In the discussion?

L316-317 According to Fig 2, rates of increase in 2018 were significantly higher than in 2013 and 2014. Other years have intermediate values. So, it is not true that rates in 2013 and 2014 were less than other years.

L318-325. Is this rate of increase different of that reported in the previous paragraph? Why is the highest in 2017. Please, clarify.

L327. The error bars and the dashed line are barely seen. Authors have to improve this. In my opinion, software limitations is not a reason. Try to delete 2011 from X-axis.

L337-338. Please rephrase

L362-368. I found this paragraph quite confusing: female biased vs. sex ratio constant vs. sex ratio declined…. Authors have to try to clarify it

Fig 4B &Fig 5. See previous comments on error bars and X-axis.

L385. Where have authors previously shown that management system does not directly affect fruit infestation?

L385-391. I think that results about insecticide use in the different management systems and the subsequent analysis will read better at the beginning of the paragraph.

L397. Change "production" to "management" in the caption and the axis to be consistent with the terminology used in L 382 .

L402-404. Delete (B). Rephrase to explain better that letters show significant differences among type of fields for males, females and total fly number separately. Additionally, analysis of total number of flies seems quite redundant if you have the analysis for females and males.

L406-407. I did not get the idea that authors have data on field size (table 1). How do they analyze it?

L481-49 0In spite of authors’ response to my comment, I still think that this should be moved to the introduction

L516-517. Please explain better here. Not in results section.

L524-525. This is already said in L515-516

L534-L536. Please, explain better

L537-545. Please explain better the discussion about relation between crop stages and fly captures

L570-573. Authors statement that fruit infestation is related to insecticide application is contradictory to what is said in L 385.

L580-582 & 621-623. According to L 384-385 management type, and therefore number of insecticides, do not have any effect on fruit infestation. Why authors conclude that more than two applications will have added benefit? Two insecticides is close to the average of high input insecticide! No need for discussing this at two points of the manuscript.

L583-589. Rephrase. It is very confusing.

Reference no. 16. Title is not correct

Author Response

Reviewer 2

I have gone through the revised version and I realized that authors did not understood the scope of my previous comments and suggestions. When in my first revision I wrote “Generally speaking, I think that authors have to work to make the M&M section more understandable and to better discuss their results”, I meant that authors should attempt a deep reorganization of the information in both sections. I probably did not explain myself well enough. Sorry about that. In the present version, mostly they have add to the text explanations to my queries, but they did not succeed to have a more concise, precise and well structured manuscript. The result is a longer manuscript (120 lines = two pages longer) that, at some points is disorganized and repetitive. I think it is really a pity because these data that really merit publication. However, authors have to critically read the manuscript and make a deep reorganization of the information. Then, this can be a very good paper. Below, there is a non- comprehensive list of my comments and suggestions. I hope they may help the authors to improve the manuscript. 

We have taken the reviewers comments to heart and attempted to reorganize the manuscript making it more concise, and hopefully more clear to readers. It has been very difficult for us to TOTALLY reorganize the paper in a drastically different way. although we have discussed this and not been able to develop a logical flow. BUT, as an alternative, we have spent much time reorganizing the Methods and Results sections in order to reduce confusion that we originally had introduced in this manuscript and hopefully clarify the flow of logical steps taken in the analysis of this study. The overall organization of the manuscript is:

1. Annual trends and phenology of adults

2. Effects of management on adults and larval fruit infestation

3. Effects of wild fruits and natural enemies

4. Action thresholds

We now state this flow in the end of the introduction as a guide to the flow in the manuscript. It has been very difficult to reduce the manuscript length because other reviewers wanted more detail and more analyses conducted.

 Specifically, we now have seven sections in the Methods that breakdown the above flow and in each section we have been careful to state the working hypotheses and the statistical approaches. The new organization is:

2.1. Data collection and analysis software 

            2.1.1 Sampling adult D. suzukii by trapping to estimate population relative          abundance 

            2.1.2. Sampling larval infestation

            2.1.3. Statistical analysis and modeling

2.2. Seasonal and annual adult D. suzukii population relative abundances 

2.3 Sex ratio changes during the study

2.4. Management system impact on D. suzukii relative abundance and fruit infestation

2.5. Crop cycle effect on D. suzukii relative abundance 

2.6. Impact of natural enemies and wild fruits on D. suzukii relative abundance in wild blueberry fields

2.7. Management - Early harvest tactic and action thresholds for D. suzukii in wild blueberry

The Results section is now aligned exactly with the proposed hypotheses and statistical approaches in the Methods. It is now organized as: 

3.1. Seasonal and annual adult D. suzukii population relative abundances

3.2 Sex ratio changes during the study

3.3. Management system impact on D. suzukii relative abundance and fruit infestation

3.4. Crop cycle effect on D. suzukii relative abundance

3.5. Impact of natural enemies and wild fruits on D. suzukii relative abundance in wild blueberry fields 

3.6. Management - Early harvest tactic and action thresholds for D. suzukii in wild blueberry

In order to match the Results better with the Methods we moved all methods of larval fruit infestation into Section 2.4 of methods - Management system impact on D. suzukii relative abundance and fruit infestation and all larval fruit infestation analyses were moved into the Results section 3.3. Management system impact on D. suzukii relative abundance and fruit infestation. 

We also broke out the sex ratioo analyses out of section 2.2. Seasonal and annual adult D. suzukii population relative abundances and put them into Section 2.3 Sex ratio changes during the study. A sex ratio section in the results is now parallel to the methods as section 3.2 Sex ratio changes during the study. This involved moving figures in the results section and renumbering them.

 We hope that this makes the manuscript much more readable.

COMMENTS

L45. A parenthesis is missing. 

We have added the missing parenthesis

L84. Add reference to Table 1. Condense all the information regarding the characterization of the 112 fields here in this paragraph, for example information in L 130-133. 

We have now condensed all of the methods fro adult trapping in the first paragraph of the Methods section.

L94, L 128, L267, etc. There is no need to repeat the sampling period unless authors feel that it will be important to precise again this information. We have deleted the repetition in line 128 that is presented in line 94. We did keep the sampling description in line 267 because it was specifically referring to the modeling of percent fruit infestation.

L96. Authors stated that trap placement was uniform for all the study except for that reported in subsection 2.6. However, in section 2.6 there is no particular information about traps deployment. Do authors refer to subsection 2.7? Again, no information about a different placement for traps is described in 2.7. The only information about different deployment of traps is in subsection 2.4 (L133-136). It is quite confusing. 

We apologize for leaving out the specific difference in trap deployment. We have now added in L96 that all traps were placed in crop fields, except in what is NOW subsection 2.5 where the traps were deployed in both crop and vegetative fields.

L128-130 and L136-138 can be merged in a single sentence. We have merged these sentences.

L169. Add order and family after Currant. This has been done.

L173-182 How was measured sex ratio? As males / (males+ females) or as males / females? Please, clarify. If both were used, use different terms to refer to these two different concepts. Sex ratio was measured as the ratio of male / (male + female) trap captures. This is now stated on line 175.

L223-226 What is different from what was described in subsection 2.1? May be it is not necessary to repeat the information.

Lines 223-226 specifically refer to the experiment described in section 2.7. This study was conducted in 2018 and was designed to test the hypothesis that wild fruits along field edges and natural enemies in fields determined the relative abundance of spotted wing drosophila. The data used for this experiment was also represented by the 2018 data of the entire data set described in subsection 2.1 and Table 1. We have now clarified this in the beginning of what is NOW subsection 2.6.

L227. Italicize D. suzukii. Thank you, we have italicized D. suzukii.

L236-237. I understood that all results from this subsection were obtained from the sub set of 20 fields (10+10). If so, this was already explained in L222-223 and there is no need to repeat it again. The reviewer is correct, the lines are redundant and we have deleted them.

L266 should read “(see methods in subsection 2.1) . We have now added the reference to the Statistical analysis and modeling section, NOW subsection 2.1.3).

L267-268. Only data of fields that were harvested before to detect Ds males in the traps were used in this analysis? Why? Clarify. 

We have clarified this...fields were only included in the model that were sampled a week prior to harvest so that we could determine the likelihood of fruit infestation the week following the recorded capture of male D. suzukii.The text now reads: "A statistical model to determine the relationship between male trap capture and percent fruit infestation the following week was fit to all wild blueberry fields that were sampled for infestation prior to harvest (n=92) between 2012 and 2017. Fields not sampled for infestation before harvest could not be used for this model even if they were monitored for adult relative abundance."

L274. Total number of fields = 112 (Table 1). So, 92 are not all the fields. We apologize for the confusion. We have rewritten the text to clarify that we are referring to a subset of the 112 fields that were harvested prior to the capture of any D. suzukii males (n=10). These fields were merely used to hypothesize that an early harvest strategy (harvest before male D. suzukii are detected in the field) might be possible. To clarify, we have rewritten the text as: "We first developed a tactic of early harvest. In all years (2012 - 2018) fields that were harvested prior to any D. suzukii males being captured were assessed as to their level of infested fruit and the date of harvest. From this data (n=10), we hypothesized that an early harvest tactic to avoiding damage was practical. To test this hypothesis, the proportion of non-ripe fruit present in 11 fields was assessed at harvest in 2016 in order to show the cost / benefit of an early harvest tactic. Three samples of 500 fruit throughout each field were collected in order to determine the mean and standard error of percent ripe fruit (green fruit / (green + ripe fruit)). A predictive linear model was developed to determine the crop loss due to non-ripe fruit and the time of harvest."

L293-294. Is this sentence complete? 

We have rewritten this sentences as:The observed infestation rates of fruit for each of the action thresholds attained the previous week were then compared to the expected probability of infested fruit derived from the probability density function of male D. suzukii action thresholds.

L300-303. In my opinion, this is not something to put in Results. In the discussion? We agree with the reviewer. We have deleted this text from the Results.

L316-317 According to Fig 2, rates of increase in 2018 were significantly higher than in 2013 and 2014. Other years have intermediate values. So, it is not true that rates in 2013 and 2014 were less than other years.

Thank you for catching this. The reviewer is correct. This was a typo...we now say "When assessed among years (Fig. 2), 2013 and 2014 rates of increase were less than 2018 which was the highest rate of increase, although not significantly different than 2012, and 2015-2017. "

L318-325. Is this rate of increase different of that reported in the previous paragraph? Why is the highest in 2017. Please, clarify. 

Yes, this rate is the daily rate of increase (cumuative) from the date of detection to the Biofix of 705 DD, whereas Figure 2 represents the daily We have clarified this by writing: 

L327. The error bars and the dashed line are barely seen. Authors have to improve this. In my opinion, software limitations is not a reason. Try to delete 2011 from X-axis. We have fixed the figure so that the line thickness of the estimated lines and standard error bars are greater. We have also covered up the 2011 and 2018 text on the x-axis.

L337-338. Please rephrase.We reworded both Figure 1 and Figure 2 captions (L 337-338) and we deleted one sentence in Figure 2 caption (line 340). 

L362-368. I found this paragraph quite confusing: female biased vs. sex ratio constant vs. sex ratio declined…. Authors have to try to clarify it

Fig 4B &Fig 5. See previous comments on error bars and X-axis. We have clarified the sex ratio presentation in the results and fixed the error bars and dashed lines in the graphs.

L385. Where have authors previously shown that management system does not directly affect fruit infestation? We had left out a sentence in this paragraph. We have added the sentence and clarified the management system effects on relative abundance of fly trap captures and fruit infestation rate. The paragraph now reads as:

We found that management system did not determine the logarithm of cumulative relative abundance of flies per trap (P > 0.05), nor the date of first fly detection. However, we have previously shown that insecticide applications affect both logarithm of cumulative relative abundance of flies per trap and fruit infestation rate. Poisson regression suggested that wild blueberry management system does determine the frequency of pesticide applications in a model that includes both year and management system (χ2(8)= 45.551, P< 0.0001). Organic systems used significantly fewer insecticide applications than the three conventional systems (low, medium, and high). Figure 6 shows the results of individual Poisson contrasts separating the number of mean pesticides applied in each management system. Therefore, management system does affect relative abundance of flies per trap and fruit infestation rate, but indirectly through insecticide application frequency which varies by management system.  

L385-391. I think that results about insecticide use in the different management systems and the subsequent analysis will read better at the beginning of the paragraph. 

Please see our restructuring of this paragraph in previous comment. We think that it is much more clear now.

L397. Change "production" to "management" in the caption and the axis to be consistent with the terminology used in L 382

Thank you. We have made these changes.

L402-404. Delete (B). Rephrase to explain better that letters show significant differences among type of fields for males, females and total fly number separately. Additionally, analysis of total number of flies seems quite redundant if you have the analysis for females and males.

The reviewer has a valid point about total flies being redundant after an analysis of males and females separately, but because many researchers and extension agents record total flies we felt that it would be informative to include total flies. We have rewritten this paragraph to read: We found that crop cycle determines adult relative abundance / trap. Significantly fewer males, females, and total flies were trapped in vegetative fields compared to fruiting fields (F(1,8)= 6.346, P = 0.036; F(1,8)= 7.803, P = 0.023; F(1,8)= 6.905, P = 0.030; for males, females, and total flies respectively). These results are shown in Figure 7.

L406-407. I did not get the idea that authors have data on field size (table 1). How do they analyze it? Field size is mentioned in the methods, but it is in the last paragraph of the methods. To clarify this we mention obtaining field size from the growers in the beginning of the methods (section 2.7. Impact of natural enemy and wild fruits on D. suzukii relative abundance in wild blueberry fields). We did state that we included field size in the general linear model in the methods: We constructed a general linear model to determine the field level factors that explained the variation in the end of season total (male + female) D. suzukii captures per trap in each field. The potential predictors that were considered for the model were: field size (ha), production system (organic, and low, medium, and high conventional), the prevalence (% of transect landcover) of non-crop wild fruit along the edges of the field, and the intensity (% of sentinel pupae predated) of predation on pupae in the field. In addition, we included all of the two and three-way interactions in the model.  

L481-49 0In spite of authors’ response to my comment, I still think that this should be moved to the introduction. 

These sentences have been removed from the Discussion.

L516-517. Please explain better here. Not in results section. 

We have taken the explanation of oscillating dynamics over time out of the results and only included them in the Discussion now. We have also fit a better model in response to another reviewer's suggestion. We have no rewritten this section to read: Our data suggest that the increase in D. suzukii trap capture each year is more likely to be oscillating, or at least stochastic in a non-linear manner and not a simple linear increase. We base this on the better balance of residulas for the non-linear model (see Fig. 1) and the better AIC index for a Fourier model compared to a linear model. This oscillating nature of increase might suggest that predation is dampening D. suzukii population increase from one year to another. We have shown that predation can reach 100% of deployed sentinel D. suzukii pupae in commercial wild blueberry fields [21]. 

L524-525. This is already said in L515-516. 

We have removed the redundancy.

L534-L536. Please, explain better

We have attempted to explain this section more clearly as follows: However, another hypothesis that might explain this oscillatory dynamic might be overwintering. An annual increase in D. suzukii relative abundance over the seven-year period is indicative of D. suzukii overwintering in Maine. There is evidence that while D. suzukii can over winter in north temperate US climate zones, survival rates are low [27-29]. Experimental studies of ours suggest that the survival rate in Maine is also low [29]. Some winters were more severe for overwintering than others during our seven-year study. The winter of 2017-2018 was one of the colder winters in recent years with extreme temperatures below -26º C in southern Maine. This was 9-10º C colder than in 2016 or 2017. Therefore, the large drop in relative abundance from 2017 to 2018 could be due to this extreme harsh winter. A negative correlation between summer relative abundance of D. suzukii and the previous winter temperatures is expected. Our analysis of relative abundance from one year to the next showed that mild winter temperatures resulted in higher relative abundance in wild blueberry fields the subsequent year and harsh winter temperatures resulted in lower relative abundance the following year. A similar finding by Briem et al. [16] provided evidence in a seven-year study in Germany that mild winters can influence relative abundance of flies.

L537-545. Please explain better the discussion about relation between crop stages and fly captures. 

We hope we clarified our discussion of crop stage as follows: Crop stage does affect relative abundance of flies as measured by the logarithm fly capture.. Fruit bearing fields were found to have significantly higher relative abundance than vegetative fields (no commercially produced fruit present). We have previously shown that wild non-blueberry fruits are important in the buildup and colonizing levels of D. suzukii in fruit bearing fields [38]. However, the existence of D. suzukii in vegetative fields also suggests that wild non-crop fruits serve as host reservoirs that can maintain populations in a field when there is no crop until the following year when a susceptible crop will be present. These vegetative field populations may also be important if flies migrate seasonally to crop field habitats. We do not know if this occurs in Maine wild blueberry, but Tait et al. [33] demonstrated that seasonal migration over large spatial scales (ca. 9 km) in Italy facilitates exploitation of resources in patchy environments. Briem et al. [16] also found that different habitats affected relative abundance of D. suzukii in Germany, but in their case, forest edges and hedgerows had higher population relative abundances than crop fields. This is opposite of what we found.  

L570-573. Authors statement that fruit infestation is related to insecticide application is contradictory to what is said in L 385.

This contradiction is due to the typo corrected in section 3.2. Management system impact on D. suzukii relative abundance. The results and Discussion are now compatible. We also changed "production to management" in the Discussion.

L580-582 & 621-623. According to L 384-385 management type, and therefore number of insecticides, do not have any effect on fruit infestation. Why authors conclude that more than two applications will have added benefit? Two insecticides is close to the average of high input insecticide! No need for discussing this at two points of the manuscript. We conclude that more than 2 insecticides will have NO benefit, but possibly we were not clear. We clarified this point because while 2 insecticides is the average for high input systems there were fields that received 3 and 4 applications. We have rewritten this section as: The average insecticide application frequency for control of D. suzukii was observed to be the highest in the high input management system, with the average insecticide application being two per season. However, some commercial fields received three and four applications during the summer. Our data suggests that more than two applications targeting D. suzukii during the growing season will have no added benefit in reducing fruit infestation.

L583-589. Rephrase. It is very confusing. Thank you for all of your suggestions. We have rewritten this section as has been stated just previously.

Reference no. 16. Title is not correct. Thank you for finding this typo. We have now inserted the correct title of the paper.
